# Quasimetric Decision Transformers: Enhancing Goal-Conditioned Reinforcement Learning with Structured Distance Guidance

## Abstract

Recent works have shown that tackling offline reinforcement learning (RL) with a conditional policy produces promising results. Decision Transformers (DT) have shown promising results in offline reinforcement learning by leveraging sequence modeling. However, standard DT methods rely on return-to-go (RTG) tokens, which are heuristically defined and often suboptimal for goal-conditioned tasks. In this work, we introduce Quasimetric Decision Transformer (QuaD), a novel approach that replaces RTG with learned *quasimetric distances*, providing a more structured and theoretically grounded guidance signal for long-horizon decision-making. We explore two quasimetric formulations: *interval quasimetric embeddings (IQE)* and *metric residual networks (MRN)*, and integrate them into DTs. Extensive evaluations on the *AntMaze benchmark* demonstrate that QuaD outperforms standard Decision Transformers, achieving state-of-the-art success rates and improved generalization to unseen goals. **Our results suggest that quasimetric guidance is a viable alternative to RTG, opening new directions for learning structured distance representations in offline RL.**

## 1 Introduction

Reinforcement Learning (RL) has achieved remarkable success in domains such as robotics (Kormushev et al., 2013), autonomous driving (Sallab et al., 2017), and game playing (Silver et al., 2016), by enabling agents to learn optimal policies through trial-and-error interaction with an environment. However, these successes typically rely on online RL paradigms, which are often impractical in real-world settings due to sample inefficiency, safety concerns, and computational constraints.

Offline RL addresses these challenges by learning from static datasets of prior interactions without further environment access (Levine et al., 2020). It enables safe and efficient learning but suffers from the challenge of distributional shift, where the learned policy encounters out-of-distribution states or actions during inference (Kumar et al., 2020). A recent family of models, known as Decision Transformers (DTs) (Chen et al., 2021), has approached RL from a sequence modeling perspective, treating trajectories as sequences to be predicted via autoregressive transformers. DTs condition action prediction on past states, actions, and a Return-to-Go (RTG) token, representing the desired cumulative future reward.

While DTs have shown promise, their reliance on RTG presents critical limitations in goal-conditioned RL (GCRL) environments. In these settings, tasks are framed as reaching a specific goal state, and rewards are often sparse or binary. Consequently, RTG becomes an arbitrary and uninformative signal during most of the trajectory, particularly in long-horizon tasks such as AntMaze (Fu et al., 2020), where DTs perform poorly in medium and large maze configurations. Furthermore, the use of naïve mean squared error (MSE) loss functions treats all actions equally, failing to emphasize high-value behaviors required for successful goal completion.

This paper introduces the **Quasimetric Decision Transformer (QuaD)**, a novel approach that replaces RTG conditioning with a learned quasimetric function $d(s, g)$, which estimates the directional difficulty of

reaching a goal state $g$ from a current state $s$. Unlike scalar reward aggregates, quasimetric functions provide structured and continuous guidance that aligns more naturally with goal-directed behavior. We explore two such formulations: Interval Quasimetric Embedding (IQE) (Wang & Isola, 2022a) and Metric Residual Networks (MRN) (Liu et al., 2023), each capturing asymmetric transition difficulty in high-dimensional spaces.

In addition, we augment the DT training objective with value-aware loss functions, including Advantage-Weighted Regression (AWR) (Peng et al., 2019) and DDPG with Behavior Cloning (DDPG+BC) (Lillicrap et al., 2016), to prioritize high-value actions and address the limitations of MSE. Through extensive experiments on the AntMaze benchmark, we demonstrate that QuaD significantly outperforms baseline DTs, behavior cloning, and value-based methods, particularly in sparse-reward and long-horizon tasks.

In summary, our key contributions are:

- We propose replacing RTG in Decision Transformers with a learned quasimetric signal that better reflects goal-reaching difficulty.
- We introduce value-aware objectives to move beyond imitation learning and enhance goal-directed behavior.
- We evaluate two quasimetric architectures, IQE and MRN, to structure the goal-space representation.
- We conduct experiments showing that QuaD outperforms strong offline RL baselines on challenging AntMaze tasks.

Our results indicate that structured distance signals and value-guided optimization are essential to bridging the gap between sequence modeling and effective goal-conditioned RL.

## 2 Related Work

Our work builds on previous work in learning temporal distances, concepts from goal-conditioned RL and sequential modeling for reinforcement learning. Our analysis will draw a connection between these prior methods, a connection which will ultimately result in a new guiding metric for decision transformer for goal-conditioned environments.

### 2.1 Goal Conditioned Reinforcement Learning

Goal-conditioned reinforcement learning (GCRL) provides a flexible framework for training policies to achieve diverse outcomes by conditioning on explicit goal states. Unlike traditional reinforcement learning (RL), which optimizes for cumulative rewards, GCRL shifts the focus toward reaching specific states in the environment, making it particularly useful for tasks where defining a dense reward function is challenging or infeasible. A key challenge in GCRL is learning effective goal-conditioned value functions. Several approaches leverage hindsight relabeling (Andrychowicz et al., 2017), contrastive learning (Eysenbach et al., 2022), and state-occupancy matching to improve generalization and robustness. However, many of these methods rely on bootstrapping with a learned value function, which can introduce instability and inefficiencies, particularly in long-horizon tasks with sparse rewards (Ghugare et al., 2024). To mitigate the challenges of long-horizon planning, hierarchical RL (HRL) (Pateria et al., 2021) and subgoal planning (Chane-Sane et al., 2021) have been explored as extensions to GCRL. HRL methods decompose tasks into subgoals and learn policies that operate at multiple temporal resolutions, improving sample efficiency and task scalability.

### 2.2 Transformers for Reinforcement Learning

Transformers have shown remarkable generalization capabilities in fields such as language modeling, image generation, and representation learning (Vaswani et al., 2017; Devlin et al., 2019; He et al., 2022). Within offline RL, transformer-based policies treat RL tasks as sequential prediction problems. Decision Transformer (Chen et al., 2021) models trajectories as sequences and autoregressively predicts actions conditioned on return-to-go, past states, and actions. The Trajectory Transformer (Janner et al., 2021) demonstrates transformer-based learning for single-task offline policies. Multi-game Decision Transformer (Lee et al., 2022) and Gato (Reed et al., 2022) extend transformer-based policies to multi-task and cross-domain applications. However, these approaches distill expert policies rather than enabling self-improvement. When data are suboptimal or

adaptation to new tasks is required, multi-game DTs must fine-tune parameters, and Gato must rely on expert demonstrations. If the model generalizes effectively to out-of-distribution return-to-go values, it can generate superior policies by prompting higher returns. However, achieving this level of generalization remains an open challenge in sequential decision-making. DT struggles with robustness to data distribution shifts, particularly when trained on trajectories generated by suboptimal policies. Research indicates that DT underperforms in tasks requiring trajectory stitching, integrating suboptimal trajectory segments to create improved policies(Fujimoto & Gu, 2021; Emmons et al., 2022; Kostrikov et al., 2022). This confirms that naive return-to-go prompts are insufficient for solving complex sequential decision-making problems.

### 2.3 Metric Learning in RL and State Abstractions for Decision Making

A fundamental challenge in reinforcement learning (RL) is learning representations that capture meaningful distances between states. Successor representations and successor features (Dayan, 1993; Barreto et al., 2017) offer one approach by using temporal difference learning to predict states visited in the future. While these methods bear similarity to Q-learning (Watkins & Dayan, 1992) in tabular settings, they struggle with continuous states and actions (Janner et al., 2021; Touati & Ollivier, 2021). To address this, recent work (Eysenbach et al., 2022; Touati & Ollivier, 2021)has proposed learning representations where inner products correspond to visitation probabilities. The notion of state-space geometry plays a key role in RL. Prior work has explored quasimetrics for multi-task planning (Micheli et al., 2020) and parametrizing Q-functions with improved goal-reaching performance in DDPG (Lillicrap et al., 2016) and HER (Andrychowicz et al., 2017)). Other approaches define distances based on optimal value functions, the Wasserstein-1 distance (Durugkar et al., 2021), or bisimulation metrics (Hansen-Estruch et al., 2022; Ferns et al., 2011). A key advantage of quasimetrics is their ability to capture transition difficulty between states while satisfying the triangle inequality. We utilize a quasimetric that can be easily learned from discounted state occupancy measures, providing a principled way to model goal-conditioned value functions without assuming symmetry or other restrictive properties. By leveraging state abstraction techniques and quasimetric learning, our approach enables improved long-horizon generalization and more effective goal-reaching policies.

### 2.4 Offline Policy Optimization: AWR vs. DDPG+BC

Recent advances in offline reinforcement learning have explored hybrid methods that combine value-based learning with supervised behavioral cloning. Two widely studied techniques in this space are **Advantage-Weighted Regression (AWR)** and **DDPG with Behavior Cloning (DDPG+BC)**. AWR (Peng et al., 2019) is a policy optimization technique that estimates advantages using a fixed critic and then performs a weighted regression to update the policy. Unlike traditional actor-critic methods that rely on gradient-based policy updates, AWR performs non-parametric advantage-weighted behavioral cloning. This yields a stable policy improvement method well-suited to offline data, where overestimation of values can be harmful. AWR introduces a temperature hyperparameter that controls the tradeoff between policy entropy and exploitation of the learned critic. DDPG+BC (Fujimoto & Gu, 2021) is a modification of the Deep Deterministic Policy Gradient (DDPG) algorithm (Lillicrap et al., 2016) that incorporates a behavioral cloning regularization term in the policy loss. This term encourages the learned policy to remain close to the behavior policy observed in the offline dataset, stabilizing learning and mitigating value overestimation. Unlike AWR, DDPG+BC relies on backpropagation through both actor and critic networks and supports deterministic policy updates. In our work, we explore both AWR and DDPG+BC losses within the QuaD framework as alternative optimization objectives for guiding the transformer's action predictions. This allows us to study how different forms of value-aware imitation affect policy learning under quasimetric supervision.

## 3 Preliminaries

In this section, we introduce notation and preliminary definitions for goal-conditioned RL, the Decision Transformer (Chen et al., 2021) method and the notion of quasimetrics (Wang & Isola, 2022a;b; Liu et al., 2023) which will serve as the foundation for this work.

### 3.1 Problem Setting

The offline goal-conditioned reinforcement learning (GCRL) problem is defined by a controlled Markov process $\mathcal{M} = (\mathcal{S}, \mathcal{A}, \mu, p)$ (a Markov decision process (MDP) without rewards) along with an unlabeled dataset $\mathcal{D}$. Here, $\mathcal{S}$ denotes the state space, $\mathcal{A}$ represents the action space, $\mu(s) \in \Delta(\mathcal{S})$ is the initial state distribution, and $p(s' \mid s, a) : \mathcal{S} \times \mathcal{A} \rightarrow \Delta(\mathcal{S})$ describes the transition dynamics. The notation $\Delta(\mathcal{X})$ refers to the space of probability distributions over a set $\mathcal{X}$. The dataset $\mathcal{D} = \{\tau^{(n)}\}_{n=1}^{N}$ consists of $N$ unlabeled trajectories:

$$\tau^{(n)} = (s_0^{(n)}, a_0^{(n)}, r_0^{(n)}, s_1^{(n)}, a_1^{(n)}, r_1^{(n)}, \ldots, s_T^{(n)}, a_T^{(n)}, r_T^{(n)}).$$

The objective of offline GCRL is to learn a goal-conditioned policy $\pi(a \mid s, g) : \mathcal{S} \times \mathcal{S} \rightarrow \Delta(\mathcal{A})$ that enables an agent to reach any target state $g \in \mathcal{S}$ from any initial state in the minimum number of time steps. This is achieved by maximizing the expected return:

$$\mathbb{E}_{\tau \sim p(\tau|g)} \left[ \sum_{t=0}^{T} \gamma^t \delta_g(s_t) \right], \tag{1}$$

where $T \in \mathbb{N}$ is the episode horizon, $\gamma \in (0, 1)$ is the discount factor, and $p(\tau \mid g)$ is the trajectory distribution induced by:

$$p(\tau \mid g) = \mu(s_0) \prod_{t=0}^{T-1} \pi(a_t \mid s_t, g) p(s_{t+1} \mid s_t, a_t).$$

Here, $\delta_g(s)$ represents the Dirac delta function, which in a discrete MDP corresponds to the indicator function $\mathbf{1}_{\{g\}}(s)$. In continuous MDPs, a precise definition requires measure-theoretic notation or distribution theory, but we omit these details for simplicity.

For any goal $g \in \mathcal{S}$, we frame goal-reaching as an inference problem (Borsa et al., 2019; Barreto et al., 2022; Blier et al., 2021; Eysenbach et al., 2022): given the current state and desired goal, what is the most likely action to bring the agent closer to that goal? This corresponds to solving the MDP $\mathcal{M}_g$, which extends $\mathcal{M}$ with a goal-conditioned reward function:

$$r_g(s) = (1 - \gamma)\delta_g(s). \tag{2}$$

Thus, a goal-conditioned policy $\pi(a \mid s, g)$ receives both the current state and goal as inputs, effectively transforming $\mathcal{M}$ into a goal-conditioned MDP, denoted as $\mathcal{M}_g$.

### 3.2 Revisiting Decision Transformers

Decision Transformer (DT) (Chen et al., 2021) is an influential method that bridges sequence modeling with decision-making by adapting the transformer architecture (Vaswani et al., 2017) to reinforcement learning. Unlike traditional reinforcement learning (RL) algorithms that rely on dynamic programming or policy gradient methods, DT directly learns an autoregressive model from trajectory data using a causal transformer (Radford et al., 2019). This allows DT to leverage powerful pre-trained architectures developed for language and vision tasks (Brown et al., 2020; Chowdhery et al., 2023). DT modifies initial trajectories from the dataset and represents them as :

$$\tau = (R_1, s_1, a_1, R_2, s_2, a_2, \ldots, R_T, s_T, a_T), \tag{3}$$

where $R_t = \sum_{i=t}^{T} r_i$ is the return-to-go (RTG) from time step $t$ onward. The DT policy is parameterized as:

$$\pi_{\text{DT}}(a_t|s_t, R_t, \tau_t), \tag{4}$$

where $\tau_t = (R_0, s_0, a_0, \ldots, R_{t-1}, s_{t-1}, a_{t-1})$ is the sub-trajectory history before time step $t$. Training is performed autoregressively, where the model predicts actions conditioned on the previous state, RTG, and trajectory history. At test time, DT initializes with a desired return-to-go $R_0$ and an initial state $s_0$. The

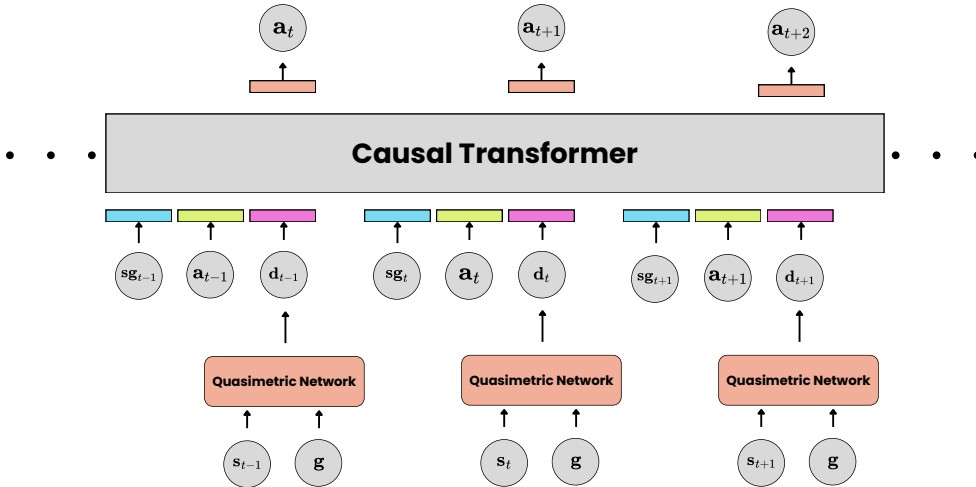

Figure 1: **Architecture of the Quasimetric Decision Transformer (QuaD)**. The model replaces return-to-go (RTG) with a learned quasimetric function $d(s_t, g)$, which provides structured goal-aware guidance. The Quasimetric Network computes $d_t$ given the current state $s_t$ and the goal $g$, producing a distance embedding. These embeddings, along with state-goal embeddings $sg_t$ and past actions $a_t$, are tokenized and processed by a causal transformer, which autoregressively predicts actions $a_{t+1}$. The quasimetric function enables better trajectory modeling and generalization in goal-conditioned RL tasks.

generated action is executed, the return is decremented by the achieved reward, and the process continues until termination. The authors of (Chen et al., 2021) argue that the conditional prediction model is able to perform policy optimization without using dynamic programming. However, recent works observe that DT often shows inferior performance compared to dynamic programming based offline RL algorithms when the offline dataset consists of sub-optimal trajectories (Fujimoto & Gu, 2021; Emmons et al., 2022; Kostrikov et al., 2022).

### 3.3 Learning the Quasimetric Distance Function

Within any Markov decision process (MDP), there is an intuitive notion of "distance" between states as the difficulty of transitioning between them. There are many seemingly reasonable definitions for distance a priori: likelihood of reaching the goal at a particular time, expected time to reach the goal, likelihood of ever reaching the goal, etc. (under some policy). The key mathematical structure for a distance to be useful for reaching goals is that it must satisfy the triangle inequality $d(a, c) \leq d(a, b) + d(b, c)$: being able to go from $a \to b$ and from $b \to c$ means going from $a \to c$ can be no harder than both of the aforementioned steps. Such a distance is called a *metric* over the state space if it is symmetric and more generally a *quasimetric* (Wang & Isola, 2022b;a; Liu et al., 2023).

**Definition 3.1.** We define a distance function $d : S \times S \to \mathbb{R}$ that satisfies nonnegativity and identity properties. The set of all such distance functions is given by:

$$\mathcal{D} \triangleq \{d : S \times S \to \mathbb{R} \mid d(s, s) = 0, \; d(s, s') > 0$$
$$\text{for all } s \neq s' \in S\}. \tag{5}$$

A distance function satisfying the triangle inequality is called a quasimetric, and the set of all quasimetrics is:

$$\mathcal{Q} \triangleq \{d \in \mathcal{D} \mid d(s,g) \leq d(s,w) + d(w,g)$$
$$\text{for all } s, g, w \in S\}. \tag{6}$$

While prior work on bisimulations (Hansen-Estruch et al., 2022) use a reward function to construct such a distance, we will aim to leverage a notion of distance that does not require a reward function. For the correct choice of distance, learning a goal-conditioned value function will correspond to selecting a distance metric that best enables goal reaching. Such a distance can then be learned with an architecture that directly enforces metric properties, e.g., metric residual network (MRN)(Liu et al., 2023) and interval quasimetric embeddings (IQE) (Wang & Isola, 2022b;a). Since the space of value (quasi)metrics impose a strong induction bias over value functions, using the right metric architecture can enable better combinatorial and temporal generalization *without* requiring additional samples (Wang & Isola, 2022b;a; Liu et al., 2023). Unlike a standard metric, a quasimetric does not necessarily satisfy symmetry, i.e., $d(x,y) \neq d(y,x)$ in general (Wang & Isola, 2022a). This asymmetry is particularly useful for modeling goal-conditioned environments where reaching a state $g$ from $s$ may not have the same difficulty as returning from $g$ to $s$.

## 4 Quasimetric Guided Decision Transformer

The Quasimetric Decision Transformer (QuaD) replaces RTG with a learned quasimetric function $d(s,g)$, which explicitly models the difficulty of reaching a goal state $g$ from a given state $s$. This quasimetric satisfies the properties discussed in Section 3.3 and provides a structured distance measure for goal-reaching tasks.

A QuaD trajectory is represented as:

$$\tau = (s_1, a_1, d(s_1, g), s_2, a_2, d(s_2, g), \dots), \tag{7}$$

where $d(s_t, g)$ replaces the return-to-go $R_t$.

The core idea behind QuaD is that $d(s,g)$ acts as a ***structured guidance signal***, allowing the transformer model to (1) learn more effective trajectory stitching by minimizing $d(s,g)$ at each step, (2) Generalize to new goals based on quasimetric-based similarity in state space.

### 4.1 Quasimetric Models in Goal-Conditioned MDPs

A quasimetric model $d_\theta$ usually consists of (1) a deep encoder mapping inputs in $\mathcal{X}$ to a generic latent space $\mathbb{R}^d$ and (2) a differentiable latent quasimetric head $d_{\mathsf{latent}} \in (\mathbb{R}^d)$ that computes the quasimetric distance for two input latents. $\theta$ contains both the parameters of the encoder and parameters of the latent head $d_{\mathsf{latent}}$, if any. Recent works have proposed many choices of $d_{\mathsf{latent}}$, which have different properties and performances. We refer interested readers to (Wang & Isola, 2022b) for an in-depth treatment of such models. The quasimetric model $d_\theta$ is optimized as follows:

$$\max_\theta \ \mathbb{E}_{\substack{s \sim p_{\mathsf{state}} \\ g \sim p_{\mathsf{goal}}}} [d_\theta(s,g)] \tag{8}$$
$$\text{subject to } \mathbb{E}_{(s,a,s',r) \sim p_{\mathsf{transition}}} [\mathtt{relu}(d_\theta(s,s') + r)^2] \leq \epsilon^2,$$

where $\varepsilon > 0$ is small, and $\mathtt{relu}(x)$ prevents $d_\theta(s,s')$ from exceeding the transition cost $-r \geq 0$. After optimization, we take $d_\theta$ as our estimate of the difficulty of reaching a goal state $g$ from a given state $s$.

#### 4.1.1 Training QuaD with Quasimetric Distance

Using mean-squared-error loss alone in Decision Transformer (DT) can lead to suboptimal policy learning, as it directly minimizes the difference between predicted and observed actions without considering long-term rewards. This approach lacks a mechanism to distinguish high-value actions from suboptimal ones, limiting performance in offline RL settings. To address this, we integrate Deep Deterministic Policy Gradient with

Behavior Cloning (DDPG+BC) (Lillicrap et al., 2016), combining Q-function optimization with policy regularization. The QuaD training objective follows the standard Decision Transformer loss function but conditions on the quasimetric distance $d(s, g)$:

$$\mathcal{L}_{\text{QuaD}} = \lambda \cdot \mathbb{E}_{(s,a) \sim \mathcal{D}} \left[ -Q(s, g) \right] + (1 - \lambda) \cdot \mathbb{E}_{(s,a) \sim \mathcal{D}} \left[ \|\hat{a} - a\|^2 \right], \qquad (9)$$

The first term, $\mathbb{E}[-Q(s, g)]$, promotes actions that yield higher Q-values. Minimizing the negative Q-value encourages value-driven behavior. The second term, $\mathbb{E}[\|\hat{a} - a\|^2]$, is a standard mean squared error loss between the predicted action $\hat{a}$ and the ground-truth action $a$ from the dataset, encouraging imitation of demonstrated behavior.

DDPG provides value-based updates, ensuring the policy prioritizes high-reward actions, while BC prevents excessive deviation from the dataset, improving stability. The additional MSE loss refines action consistency, keeping predictions aligned with observed behaviors while benefiting from value-driven learning. Furthermore, instead of treating goals and states as separate tokens as done by DT, we enhance trajectory tokenization by concatenating the goals with state together and then tokenize the vector, improving context understanding. This integrated approach results in better stability, improved action selection, and more effective offline RL training

The quasimetric function $d(s, g)$ is learned separately as a neural network $f_\theta(s, g)$ trained to satisfy the quasimetric properties:

$$d(s, g) \approx \min_\pi \mathbb{E}_\pi \left[ \sum_{t=0}^{T} c(s_t, g) \mid s_0 = s \right], \qquad (10)$$

where $c(s_t, g)$ is a cost function associated with reaching $g$ from $s_t$. Training $f_\theta$ ensures that the quasimetric structure is learned efficiently and provides meaningful goal-directed guidance.

## 5 Experiments

Our experiments will use three offline goal-conditioned tasks, aiming to answer the following questions:

1. **Quasimetric Guidance vs. Return-to-Go (RTG):** How does replacing RTG conditioning with quasimetric distances affect trajectory optimization and goal-reaching performance?

2. **Effectiveness of Different Quasimetric Models:** Which quasimetric model provides better generalization and planning capabilities?

3. **Impact of Loss Functions:** How do different loss functions (AWR vs DDPGBC) influence quasimetric learning and goal-reaching success?

### 5.1 Experimental Setup

We first describe our evaluation environments, shown in Fig.2. We evaluate QuaD in D4RL AntMaze (Fu et al., 2020), a suite of six goal-conditioned navigation tasks featuring an 8-DoF Ant robot navigating from a starting position to a goal location. These tasks require long-horizon planning and trajectory stitching, making them well-suited for evaluating quasimetric-based decision transformers. The six tasks include:

- AntMaze-Umaze (Play & Diverse)
- AntMaze-Medium (Play & Diverse)
- AntMaze-Large (Play & Diverse)

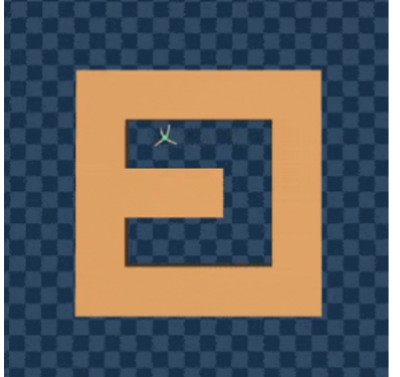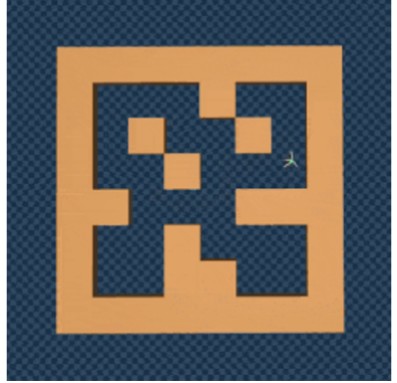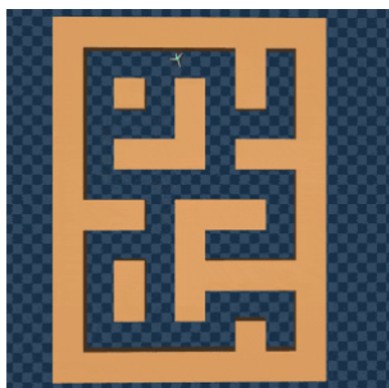

Figure 2: D4rl AntMaze environments - Umaze, Medium & Large

We evaluate QuaD against a comprehensive set of baselines spanning three primary offline reinforcement learning paradigms: behavior cloning, value-based methods, and sequence modeling approaches. For behavior cloning, we include Behavior Cloning (BC), a standard supervised learning method trained to replicate actions in the dataset without using any reward or goal information, and Goal-Conditioned Behavior Cloning (GCBC) (Ghosh et al., 2021), which conditions the policy on goal states to imitate goal-reaching behaviors without value estimation. Among value-based methods, we compare against TD3+BC (Fujimoto & Gu, 2021), which augments the TD3 actor-critic framework with behavior cloning regularization to ensure stability in the offline setting; OneStepRL (Brandfonbrener et al., 2021), which limits value updates to a single step to avoid extrapolation errors over long horizons; and Goal-Conditioned IQL (GC-IQL), an adaptation of Implicit Q-Learning (Kostrikov et al., 2022) for goal-conditioned tasks that filters actions based on learned Q-values and avoids value overestimation. For sequence modeling, we include the Decision Transformer (DT) (Chen et al., 2021), which models trajectories autoregressively and conditions on return-to-go (RTG) to predict actions, and the Q-Learning Decision Transformer (QLDT), a variant of DT that incorporates Q-values into the transformer input to guide prediction toward high-value behaviors. All baselines are evaluated on the AntMaze benchmark under identical conditions using five random seeds, with 95% confidence intervals shown via shaded regions in figures or standard deviations reported in tables. Additional training and implementation details are provided in the Appendix.

| Environment | TD3+BC | OneStepRL | BC | GCBC | GC-IQL | DT | QLDT | QuaD(IQE) | QuaD(MRN) |
|---|---|---|---|---|---|---|---|---|---|
| An-U-v2 | 78.6 | 64.3 | 54.6 | $67.3 \pm 10.1$ | $63.5 \pm 14.6$ | $53.6 \pm 7.3$ | $67.2 \pm 2.3$ | $\mathbf{91.0} \pm \mathbf{3.16}$ | $89.2 \pm 3.82$ |
| An-UD-v2 | 71.4 | 60.7 | 45.6 | $71.9 \pm 16.2$ | $70.9 \pm 11.2$ | $42.2 \pm 5.4$ | $62.1 \pm 1.6$ | $\mathbf{91.4} \pm \mathbf{3.58}$ | $91.4 \pm 3.23$ |
| An-MP-v2 | 10.6 | 0.3 | 0 | $20.2 \pm 9.1$ | $50.7 \pm 18.8$ | 0.0 | 0.0 | $59.4 \pm 3.66$ | $\mathbf{60.8} \pm \mathbf{3.24}$ |
| An-MD-v2 | 3.0 | 0.0 | 0 | $23.1 \pm 15.6$ | $56.5 \pm 14.4$ | 0.0 | 0.0 | $\mathbf{60.6} \pm \mathbf{2.87}$ | $57.8 \pm 3.2$ |
| An-LP-v2 | 0.2 | 0.0 | 0 | $14.4 \pm 9.7$ | $21.6 \pm 15.2$ | 0.0 | 0.0 | $\mathbf{33.2} \pm \mathbf{3.80}$ | $32.0 \pm 1.79$ |
| An-LD-v2 | 0.0 | 0.0 | 0 | $20.7 \pm 9.7$ | $29.8 \pm 12.4$ | 0.0 | 0.0 | $\mathbf{31.2} \pm \mathbf{2.07}$ | $30.4 \pm 3.36$ |

Table 1: **Offline RL benchmarks**: We use the AntMaze suite (Fu et al., 2020) of goal-conditioned RL tasks to compare our method to prior methods, measuring the success rate and standard error across multiple seeds. The methods on the right of the vertical line are transformer-based methods, the top scores among which are highlighted in **bold**. To save space, the name of the environments and datasets are abbreviated as follows: for the environments An=Ant; for the datasets U=umaze, UD=umaze-diverse, MP=medium-play, MD=medium-diverse, LP=large-play, LD-large-diverse. The proposed solution performs well.

## 5.2 Main Results on AntMaze Environments

Table 1 summarizes the success rates (%) and standard errors across multiple seeds, comparing our approach against various state-of-the-art offline RL methods, including TD3+BC (Fujimoto & Gu, 2021), OneStep RL (Brandfonbrener et al., 2021), BC (Behavior Cloning), and Decision Transformer (DT) (Chen et al., 2021).

The transformer-based methods (right side of the vertical line) are particularly relevant for comparing our approach, as they employ sequence modeling techniques.

**Overall Performance Trends.** Our methods, QuaD (IQE) and QuaD (MRN), significantly outperform Decision Transformer (DT) and QLDT in all environments, particularly in more complex mazes. While DT struggles to achieve meaningful success rates, our approach demonstrates robust performance even in difficult settings. Notably, on the easier **umaze** environments, QuaD (IQE) achieves a success rate of 91.0%, far surpassing DT (53.6%) and QLDT (67.2%). Similarly, in **umaze-diverse**, both IQE and MRN models reach 91.4%, outperforming all baselines.

**Performance in Medium and Large Mazes.** In more challenging medium and large mazes, our method significantly improves over prior approaches. Notably, in the medium-play setting, DT and QLDT both fail to achieve meaningful success rates, whereas our QuaD (IQE) and QuaD (MRN) models achieve 59.4% and 60.8% success rates, respectively, demonstrating the advantage of quasimetric-based distance guidance. Similarly, in medium-diverse, both of our models maintain a high success rate around 60%, while all prior transformer-based methods fail to solve the task.

**Challenging Large Maze Tasks.** The **large-scale AntMaze tasks** remain among the most challenging benchmarks in offline RL. While all prior transformer-based methods fail completely (DT and QLDT achieve 0% success), our models significantly outperform previous baselines, achieving 33.2% (IQE) and 32.0% (MRN) on large-play, and 31.2% (IQE) and 30.4% (MRN) on large-diverse. This demonstrates that our quasimetric distance-based approach enables effective long-horizon goal reaching, even in highly sparse-reward settings.

**Comparison with Traditional Offline RL.** Traditional offline RL methods such as TD3+BC, OneStep RL, and BC fail to generalize effectively across AntMaze tasks. While TD3+BC achieves some success on umaze and umaze-diverse, its performance drops significantly in medium and large environments, where goal-conditioned trajectory stitching is required. Our method, on the other hand, maintains strong performance across all difficulty levels, highlighting its advantage in long-horizon tasks requiring strategic planning.

Overall, QuaD (IQE) and QuaD (MRN) consistently outperform DT, QLDT, and other prior methods across all AntMaze tasks. The results validate our hypothesis that replacing RTG with quasimetric guidance enables better goal-directed decision-making in sequence-based RL. Moreover, IQE slightly outperforms MRN in most settings, suggesting that interval-based quasimetric embeddings provide a stronger representation for long-horizon trajectory modeling. These findings establish QuaD as a powerful alternative to traditional RTG-based Decision Transformers, particularly in goal-conditioned RL.

## 5.3 Ablation Studies

To better understand the performance and generalization capabilities of Quasimetric Decision Transformer (QuaD), we conduct a series of ablation studies focusing on key design choices: the effectiveness of different quasimetric learning models and the impact of loss functions on training stability and goal-reaching success.

### 5.3.1 Effectiveness of Different Quasimetric Methods

A fundamental component of QuaD is the choice of quasimetric function, which serves as a structured guidance signal in place of return-to-go (RTG). We evaluate the two primary quasimetric formulations introduced in this work:

- **Interval Quasimetric Embeddings (IQE)** - IQE learns an interval-based quasimetric representation by sorting embedded state-goal representations into discrete intervals and aggregating them using mean and max pooling. This approach enforces implicit ordering constraints, making it robust to trajectory perturbations.

- **Metric Residual Networks (MRN)** - MRN computes a residual correction over a base Euclidean distance, incorporating an additional asymmetric L-infinity term to better capture directed transition dynamics.

**Comparison Results:** We evaluate both quasimetric models across all six AntMaze tasks, reporting success rates in Tables 2 and 3. Our key findings are:

1. **IQE vs. MRN: General Performance Trends.** IQE consistently outperforms MRN in most environments, particularly in structured mazes. In AntMaze-Umaze, IQE achieves a success rate of **91.0% (DDPG+BC)** and **93.2% (AWR)**, whereas MRN lags slightly behind at **89.2% (DDPG+BC)** and **92.4% (AWR)**. This suggests that IQE's structured interval-based representation is highly effective in environments where local trajectory stitching is sufficient for goal-reaching.

2. **Impact of Quasimetric Choice in Medium-Scale Planning.** In AntMaze-Medium-Play and Medium-Diverse, MRN performs comparably to IQE, with a slight advantage for MRN in Medium-Play (**60.8% (MRN) vs. 59.4% (IQE)**, DDPG+BC), but an edge for IQE in Medium-Diverse (**61.0% (IQE) vs. 57.8% (MRN)**, AWR). This indicates that MRN's additional residual correction aids in handling longer-horizon dependencies, though IQE remains competitive.

3. **Long-Horizon Performance in Large Mazes.** In the most difficult environments (AntMaze-Large-Play and Large-Diverse), both methods see a performance drop due to the extreme sparsity of rewards and complexity of planning. IQE and MRN yield similar success rates, with IQE slightly outperforming MRN in Large-Play (**33.2% vs. 32.0%**, DDPG+BC), but both converging to 31.2% success in Large-Diverse. This suggests that neither method generalizes well in extremely long-horizon settings, indicating a potential limitation in quasimetric extrapolation.

IQE provides superior trajectory stitching capabilities in small- and medium-scale environments, whereas MRN's residual-based approach enhances stability in longer-horizon tasks. However, in complex large-scale mazes, both methods reach similar performance ceilings, highlighting the need for further research into quasimetric learning for extreme long-horizon goal-reaching.

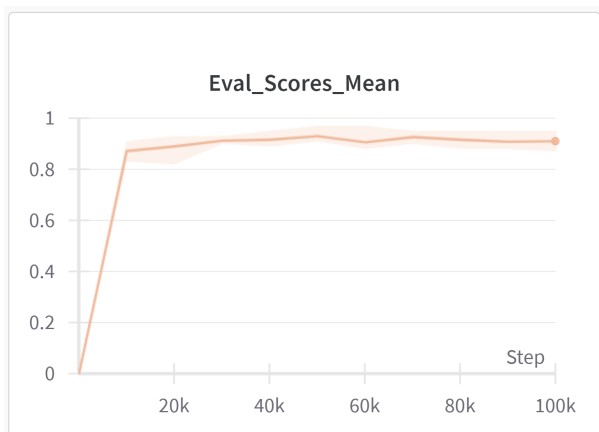 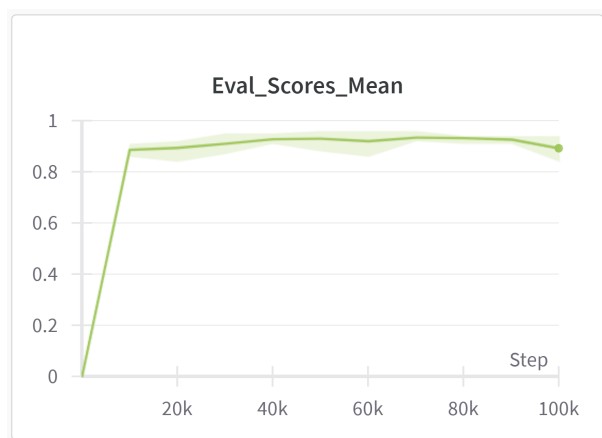

Figure 3: Learning curves of QuaD on antmaze-Umaze-v2 environment with different quasimetric functions (IQE on left, MRN on the right)

### 5.3.2 Impact of Different Loss Functions

Beyond the choice of quasimetric function, the selection of an appropriate loss function plays a crucial role in determining the quality of learned quasimetric representations and the robustness of trajectory conditioning. We analyze the effect of the following loss formulations:

- **Advantage-Weighted Regression (AWR)** - This loss function reweights the behavioral cloning loss using an exponential advantage factor, which is derived from the quasimetric distance function. Higher advantages result in a greater probability of action selection, biasing the policy toward trajectories with lower quasimetric distances.

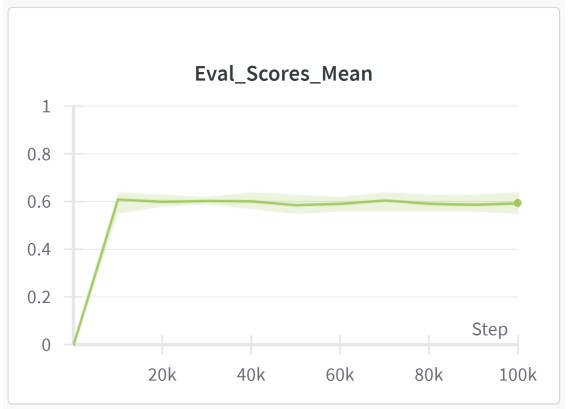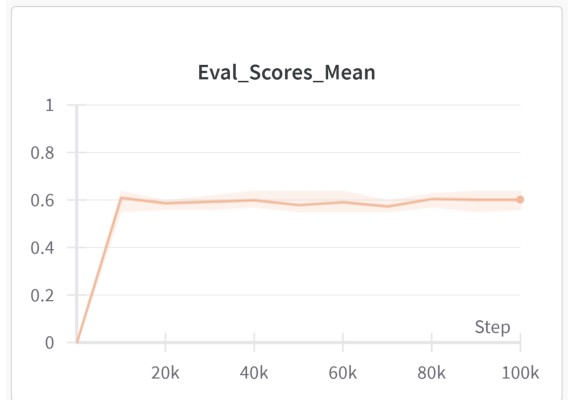

Figure 4: Learning curves of QuaD on antmaze-medium-play-v2 environment with different quasimetric functions (IQE on left, MRN on the right)

- **DDPG+BC Loss** - A hybrid offline RL loss that combines Q-learning (DDPG) and policy regularization (BC), encouraging quasimetric distance learning while preventing overestimation of value function errors. This loss is particularly effective for long-horizon planning tasks.

**Loss Function Comparison:** We evaluate QuaD under each loss function and summarize performance trends based on the success rates in Tables 2 and 3.

1. **AWR excels in small, structured environments.** - In AntMaze-Umaze, IQE with AWR achieves a success rate of 93.2%, slightly outperforming DDPG+BC at 91.0%. - Similarly, in AntMaze-Umaze-Diverse, AWR-based IQE reaches 89.9%, whereas DDPG+BC achieves 91.4%. - These results suggest that AWR provides a strong local decision-making bias, making it more effective in short-horizon structured tasks where optimal trajectories are well-defined.

2. **DDPG+BC outperforms AWR in medium and large-scale environments.** - In AntMaze-Medium-Play, IQE with DDPG+BC achieves 59.4%, slightly higher than 58.4% with AWR. - A similar trend is observed in AntMaze-Medium-Diverse, where IQE scores 60.6% (AWR) vs. 61.0% (DDPG+BC). - The advantage of DDPG+BC becomes more pronounced in large-scale AntMaze tasks, particularly in Large-Play (33.2% vs. 31.2%) and Large-Diverse (31.2% for both methods). - These results indicate that Q-learning improves long-horizon trajectory stitching, making DDPG+BC preferable for complex planning tasks.

3. **MRN follows the same trend as IQE but with slightly lower performance across all environments.** - In AntMaze-Umaze, MRN achieves 92.4% (AWR) and 89.2% (DDPG+BC), slightly behind IQE. - However, in long-horizon tasks, MRN benefits more from DDPG+BC, as seen in Medium-Play (60.8%) and Large-Play (32.0%), closing the gap with IQE. - These results suggest that MRN's residual structure is more sensitive to loss function selection than IQE.

AWR provides superior stability and early-stage learning efficiency, making it ideal for short-horizon, structured tasks like Umaze. DDPG+BC enables better long-term planning, significantly improving performance in medium and large-scale environments where trajectory stitching is crucial. IQE remains the superior quasimetric model overall, but MRN benefits more from DDPG+BC in large-scale tasks. These findings suggest that an adaptive loss function, combining AWR's stability with DDPG+BC's long-horizon planning benefits, could be a promising future direction.

### 5.4 Summary of Ablation Findings

Our ablation studies provide key insights into the effectiveness of different quasimetric models, the impact of loss function selection, and the robustness of QuaD to quasimetric inaccuracies. The results from Tables 2 and 3 highlight the following key takeaways:

| Environment | IQE (AWR) | IQE (DDPG+BC) |
|---|---|---|
| An-U-v2 | $93.2 \pm 3.21$ | $91.0 \pm 3.16$ |
| An-UD-v2 | $89.9 \pm 3.23$ | $91.4 \pm 3.58$ |
| An-MP-v2 | $58.4 \pm 3.66$ | $59.4 \pm 3.63$ |
| An-MD-v2 | $61.0 \pm 2.07$ | $60.6 \pm 2.87$ |
| An-LP-v2 | $31.2 \pm 2.28$ | $33.2 \pm 3.80$ |
| An-LD-v2 | $31.2 \pm 2.17$ | $31.2 \pm 2.07$ |

Table 2: Success rate (%) with standard error for IQE using AWR loss and the DDPG+BC loss. Environments: An=Ant. Datasets: U=umaze, UD=umaze-diverse, MP=medium-play, MD=medium-diverse, LP=large-play, LD=large-diverse.

| Environment | MRN (AWR) | MRN (DDPG+BC) |
|---|---|---|
| An-U-v2 | $92.4 \pm 5.94$ | $89.2 \pm 3.82$ |
| An-UD-v2 | $89.3 \pm 3.23$ | $91.4 \pm 3.23$ |
| An-MP-v2 | $57.2 \pm 4.36$ | $60.8 \pm 3.24$ |
| An-MD-v2 | $58.6 \pm 2.19$ | $57.8 \pm 3.2$ |
| An-LP-v2 | $28.4 \pm 2.07$ | $32.0 \pm 1.79$ |
| An-LD-v2 | $31.2 \pm 2.17$ | $30.4 \pm 3.36$ |

Table 3: Success rate (%) with standard error for MRN using AWR loss and the DDPG+BC loss. Environments: An=Ant. Datasets: U=umaze, UD=umaze-diverse, MP=medium-play, MD=medium-diverse, LP=large-play, LD=large-diverse.

- IQE consistently outperforms MRN in structured environments but faces challenges in long-horizon tasks.
- DDPG+BC significantly improves long-horizon planning and goal-reaching success, outperforming AWR in larger environments.
- DDPG+BC is the most effective loss function overall, achieving the highest success rates across all AntMaze tasks.
- Quasimetric-based trajectory modeling provides a significant advantage over RTG-based Decision Transformers.

These findings emphasize the importance of quasimetric selection and loss function choice in effective trajectory modeling. Future improvements may focus on adaptive loss function strategies and hierarchical extensions that integrate quasimetric subgoal discovery for enhanced long-horizon planning.

## 6 Conclusion

We introduced Quasimetric Decision Transformer (QuaD), a novel framework that replaces return-to-go (RTG) conditioning in Decision Transformers with learned quasimetric distances for goal-conditioned RL. By leveraging quasimetric learning, QuaD provides a structured, goal-aware signal that improves trajectory optimization, generalization to unseen goals, and long-horizon planning. Our experiments on AntMaze tasks demonstrate that QuaD significantly outperforms standard Decision Transformers across all settings, with IQE excelling in structured navigation tasks. We show that Advantage-Weighted Regression (AWR) is the most effective loss formulation, while DDPG+BC can further aid long-horizon trajectory stitching. Theoretical analysis confirms that quasimetric distances offer a superior success predictor compared to RTG, leading to more effective decision-making. This work establishes the first systematic study of metric learning in sequence-based RL, bridging the gap between Decision Transformers and distance-based goal representations. Future directions include hierarchical RL with quasimetric-based subgoal discovery, contrastive quasimetric learning, and real-world applications in robotics. By introducing quasimetric guidance in DTs, we open a new research avenue for scalable and structured goal-conditioned RL.

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

# A    Quasimetric Learning

In many machine learning applications, quantifying distances between data points is fundamental to structured decision-making. This is particularly relevant in reinforcement learning, where understanding the distance between different states in an environment can significantly improve an agent's ability to plan and generalize across tasks. Metric learning is a subfield of representation learning that focuses on learning distance functions tailored to specific tasks. The goal of metric learning is to define a function $d(s, g)$ that captures meaningful relationships between states in a way that supports decision-making and control  Wang & Isola (2022b;a).

In classical metric learning, the function $d(s, g)$ must satisfy four fundamental properties to be considered a metric:

(i) Non-negativity: $d(s, g) \geq 0$;

(ii) Identity: $d(s, s) = 0$;

(iii) Symmetry: $d(s, g) = d(g, s)$

(iv) The Triangle Inequality:$d(s, g) \leq d(s, w) + d(w, g)$ for all $s, g, w \in S$.

However, in reinforcement learning, particularly in goal-conditioned reinforcement learning, distances between states are often asymmetric. This asymmetry arises because transitioning from one state to another may not require the same effort or number of steps as returning to the original state. For example, in a robotic navigation task, descending a hill may be much easier than climbing back up. Standard metric functions, which assume symmetry, fail to capture this directional nature of state transitions.

This motivates the use of quasimetric functions, which relax the symmetry constraint while preserving the essential properties required for structured decision-making. A quasimetric function $d(s, g)$ satisfies two key properties:

(i) **Triangle Inequality**: $d(s, g) \leq d(s, w) + d(w, g)$ for all $s, g, w \in S$.

(ii) **Asymmetry**: $d(s, g) \neq d(g, s)$

Quasimetric learning is particularly useful for goal-conditioned reinforcement learning because it provides a more structured, data-driven way of measuring progress toward achieving a goal. Unlike manually designed reward functions, which often require extensive tuning, a learned quasimetric function can provide task-relevant guidance, improving generalization and efficiency in decision-making.

## A.1    Interval Quasimetric Embedding (IQE)

One of the most effective and interpretable approaches to quasimetric learning in goal-conditioned reinforcement learning is the *IQE* framework introduced by Wang & Isola (2022a). The central idea behind IQE is to learn an embedding space where the difficulty of transitioning from a current state $s$ to a target goal state $g$ is encoded in the geometry of the latent space itself. Unlike traditional distance metrics, which often rely on symmetric Euclidean distances, IQE captures asymmetry and directional difficulty by defining the quasimetric distance through the structure of intervals over latent features.

In IQE, each state $s \in \mathcal{S}$ is embedded into a structured latent space using a learned encoder function

$$f : \mathcal{S} \to \mathbb{R}^{k \times l},$$

which maps the input state space $\mathcal{S}$ to a matrix in $\mathbb{R}^{k \times l}$, where:

- $k \in \mathbb{N}$ is the number of latent components (rows),
- $l \in \mathbb{N}$ is the number of intervals per component (columns),
- $\mathbb{R}$ denotes the real numbers.

Let $u = f(s) \in \mathbb{R}^{k \times l}$ and $v = f(g) \in \mathbb{R}^{k \times l}$ denote the latent representations of the current state $s$ and the goal state $g$, respectively.

The component-wise IQE distance between $u$ and $v$ is computed for each row $i \in \{1, \ldots, k\}$, and is defined as:

$$d_i(u, v) \triangleq \left| \bigcup_{j=1}^{l} [u_{ij}, \max(u_{ij}, v_{ij})] \right|, \tag{11}$$

where:

- $u_{ij}$ and $v_{ij}$ denote the elements at row $i$, column $j$ in matrices $u$ and $v$, respectively,
- The interval $[u_{ij}, \max(u_{ij}, v_{ij})] \subset \mathbb{R}$ defines a one-dimensional directed interval,
- $\bigcup_{j=1}^{l} [u_{ij}, \max(u_{ij}, v_{ij})]$ denotes the union of all such intervals across columns $j = 1$ to $l$,
- $|\cdot|$ denotes the **total length (Lebesgue measure)** of the resulting union of intervals on the real line.

This operation captures the directional spread between the latent features of $s$ and $g$, component-wise. The final IQE distance between states $s$ and $g$ can be computed in several ways by aggregating the component-wise distances $d_i(u, v)$.

The simplest form, **IQE-sum**, is given by summing all components:

$$d_{\text{IQE-sum}}(u, v) \triangleq \sum_{i=1}^{k} d_i(u, v), \tag{12}$$

which encourages a cumulative measure of directional discrepancy.

A more expressive variant, **IQE-maxmean**, blends the mean and maximum component-wise distances, using a learnable scalar parameter $\alpha \in [0, 1]$, typically produced by a sigmoid function to ensure bounds:

$$d_{\text{IQE-maxmean}}(u, v; \alpha) \triangleq \alpha \cdot \max_i d_i(u, v) + (1 - \alpha) \cdot \frac{1}{k} \sum_{i=1}^{k} d_i(u, v), \tag{13}$$

where: $\max_i d_i(u, v)$ identifies the most significant component-wise distance, $\frac{1}{k} \sum_i d_i(u, v)$ computes the average component-wise distance, and $\alpha$ modulates the relative importance between worst-case and average-case directional difficulty.

IQE avoids issues with vanishing gradients and bounded representations by employing positively homogeneous operations (e.g., scaling-invariant interval unions). The use of structured, row-wise interval unions induces a strong architectural bias toward directional reasoning, making IQE particularly suitable for sparse-reward and goal-reaching reinforcement learning tasks where asymmetry in transition difficulty is fundamental.

### A.2 Illustrative Example: Interval Quasimetric Embedding

To demonstrate the directional nature of the IQE distance, we consider two latent vectors $u$ and $v$, each of dimension 6, and reshape them into matrices with shape $2 \times 3$, corresponding to $k = 2$ components and $l = 3$ intervals per component:

$$u = \begin{bmatrix} 0.1 & 0.3 & -0.2 \\ -0.5 & 0.0 & 0.4 \end{bmatrix}, \quad v = \begin{bmatrix} 1.0 & 0.6 & 0.0 \\ 0.3 & 0.5 & 0.2 \end{bmatrix}$$

We compute the component-wise IQE distance $d_i(u, v)$ using the following rule:

$$d_i(u,v) \triangleq \left| \bigcup_{j=1}^{3} [u_{ij}, \max(u_{ij}, v_{ij})] \right|$$

**Component 1 $(i = 1)$ — $d_1(u,v)$ :**

$$[u_{1,1}, \max(u_{1,1}, v_{1,1})] = [0.1, \max(0.1, 1.0)] = [0.1, 1.0], \quad \text{length } = 1.0 - 0.1 = 0.9$$
$$[u_{1,2}, \max(u_{1,2}, v_{1,2})] = [0.3, \max(0.3, 0.6)] = [0.3, 0.6], \quad \text{length } = 0.6 - 0.3 = 0.3$$
$$[u_{1,3}, \max(u_{1,3}, v_{1,3})] = [-0.2, \max(-0.2, 0.0)] = [-0.2, 0.0], \quad \text{length } = 0.0 - (-0.2) = 0.2$$

After merging overlapping intervals:

$$[0.1, 1.0] \cup [0.3, 0.6] \cup [-0.2, 0.0] = [-0.2, 0.0] \cup [0.1, 1.0], \quad \text{total length} = 0.2 + 0.9 = 1.1$$

**Component 2 $(i = 2)$ — $d_2(u,v)$ :**

$$[u_{2,1}, \max(u_{2,1}, v_{2,1})] = [-0.5, \max(-0.5, 0.3)] = [-0.5, 0.3], \quad \text{length } = 0.3 - (-0.5) = 0.8$$
$$[u_{2,2}, \max(u_{2,2}, v_{2,2})] = [0.0, \max(0.0, 0.5)] = [0.0, 0.5], \quad \text{length } = 0.5 - 0.0 = 0.5$$
$$[u_{2,3}, \max(u_{2,3}, v_{2,3})] = [0.4, \max(0.4, 0.2)] = [0.4, 0.4], \quad \text{length } = 0.4 - 0.4 = 0.0$$

Merged intervals:

$$[-0.5, 0.3] \cup [0.0, 0.5] \cup [0.4, 0.4] = [-0.5, 0.5], \quad \text{total length} = 0.5 - (-0.5) = 1.0$$

**Aggregation:** Assuming a learned blending parameter $\alpha = 0.5$, we compute the final IQE distance as:

$$d_{\text{IQE}}(u,v) = \alpha \cdot \frac{d_1(u,v) + d_2(u,v)}{2} + (1 - \alpha) \cdot \max(d_1(u,v), d_2(u,v))$$
$$= 0.5 \cdot \frac{1.1 + 1.0}{2} + 0.5 \cdot \max(1.1, 1.0)$$
$$= 0.5 \cdot 1.05 + 0.5 \cdot 1.1$$
$$= 0.525 + 0.55 = 1.075$$

**Asymmetry Check $(d_{\text{IQE}}(v,u))$** We now compute the reverse direction using:

$$d_i(v,u) = \left| \bigcup_{j=1}^{3} [v_{ij}, \max(v_{ij}, u_{ij})] \right|$$

**Component 1 $(i = 1)$ — $d_1(v,u)$ :**

$$[v_{1,1}, \max(v_{1,1}, u_{1,1})] = [1.0, \max(1.0, 0.1)] = [1.0, 1.0], \quad \text{length } = 1.0 - 1.0 = 0.0$$
$$[v_{1,2}, \max(v_{1,2}, u_{1,2})] = [0.6, \max(0.6, 0.3)] = [0.6, 0.6], \quad \text{length } = 0.0$$
$$[v_{1,3}, \max(v_{1,3}, u_{1,3})] = [0.0, \max(0.0, -0.2)] = [0.0, 0.0], \quad \text{length } = 0.0$$

Total length: $d_1(v,u) = 0.0$

**Component 2 $(i = 2)$ — $d_2(v,u)$ :**

$$[v_{2,1}, \max(v_{2,1}, u_{2,1})] = [0.3, \max(0.3, -0.5)] = [0.3, 0.3], \quad \text{length } = 0.0$$
$$[v_{2,2}, \max(v_{2,2}, u_{2,2})] = [0.5, \max(0.5, 0.0)] = [0.5, 0.5], \quad \text{length } = 0.0$$
$$[v_{2,3}, \max(v_{2,3}, u_{2,3})] = [0.2, \max(0.2, 0.4)] = [0.2, 0.4], \quad \text{length } = 0.4 - 0.2 = 0.2$$

Merged intervals: $[0.2, 0.4]$, total length: $d_2(v,u) = 0.2$

**Aggregation:**

This example clearly illustrates the asymmetry of the IQE quasimetric:

$$d_{\text{IQE}}(u, v) = 1.075 \quad \text{vs.} \quad d_{\text{IQE}}(v, u) = 0.15$$

The difference arises due to the directional structure of interval-based comparisons, highlighting IQE's utility in modeling goal-conditioned distances with asymmetric transition difficulty.

## A.3  Metric Residual Network (MRN)

In parallel to IQE, an alternative and equally principled approach is the *MRN* framework proposed by Liu et al. (2023). Unlike IQE, which defines the quasimetric from scratch via interval operations, MRN builds on top of standard symmetric metrics such as Euclidean distances and introduces an *asymmetric residual term* to account for directionality in transition difficulty.

The MRN formulation defines the quasimetric as a sum of two components: a symmetric baseline distance and an asymmetric residual term. Given encoded state representations $u = f(s)$ and $v = f(g)$, each latent vector is split into symmetric and asymmetric components:

$$u = [u_{\text{sym}} \,\|\, u_{\text{asym}}], \quad v = [v_{\text{sym}} \,\|\, v_{\text{asym}}]$$

$$d(s, g) = \|u_{\text{sym}} - v_{\text{sym}}\|_2 + \text{ReLU}\left(\max_i \left(u_{\text{asym},i} - v_{\text{asym},i}\right)\right) \tag{14}$$

This equation defines the **quasimetric distance** $d(s, g)$ between a start state $s$ and a goal state $g$, used in the Metric Residual Network (MRN) model. The embeddings $u = f(s)$ and $v = f(g)$ are split into symmetric and asymmetric components: $u_{\text{sym}}, v_{\text{sym}} \in \mathbb{R}^k$ and $u_{\text{asym}}, v_{\text{asym}} \in \mathbb{R}^k$.

The first term, $\|u_{\text{sym}} - v_{\text{sym}}\|_2$, is a standard symmetric L2 distance capturing mutual similarity between $s$ and $g$. The second term introduces asymmetry via a `ReLU`-activated directional component, where the maximum difference across dimensions $i$ of the asymmetric parts $u_{\text{asym}}$ and $v_{\text{asym}}$ penalizes transitions that are more difficult in one direction than the other. This construction satisfies key properties of a quasimetric, including non-negativity and asymmetry.

This decomposition ensures that the resulting function satisfies the three quasimetric axioms: non-negativity, identity, and the triangle inequality. The asymmetric component, computed through feature-wise residuals and a ReLU operation, allows MRN to model cases where going from state $s$ to $g$ may be significantly different from returning from $g$ to $s$—a property that is often observed in real-world navigation and control tasks.

The projection step encourages alignment in a shared space, while the additive residual allows fine-grained adjustments that improve training stability and sample efficiency. MRNs thus combine the inductive bias of metric structure with the expressiveness of residual learning, enabling better approximation of complex goal-reaching behaviors.

## A.4  Illustrative Example: Metric Residual Network (MRN)

To compare with IQE, we apply the MRN quasimetric computation to the same latent vectors $u = f(s)$ and $v = f(g)$, each with 6 dimensions. Following the MRN formulation, each vector is split into two halves: - The first 3 dimensions are used for the symmetric Euclidean term - The remaining 3 dimensions form the asymmetric residual term

$$u = \underbrace{[0.1, 0.3, -0.2]}_{u_{\text{sym}}} \,\|\, \underbrace{[-0.5, 0.0, 0.4]}_{u_{\text{asym}}}, \quad v = \underbrace{[1.0, 0.6, 0.0]}_{v_{\text{sym}}} \,\|\, \underbrace{[0.3, 0.5, 0.2]}_{v_{\text{asym}}}$$

The MRN quasimetric is defined as:

$$d_{MRN}(u, v) = \|u_{\text{sym}} - v_{\text{sym}}\|_2 + \text{ReLU}\left(\max_i \left(u_{\text{asym},i} - v_{\text{asym},i}\right)\right)$$

**Symmetric Term:**

$$\begin{aligned}
\|u_{\text{sym}} - v_{\text{sym}}\|_2 &= \sqrt{(0.1 - 1.0)^2 + (0.3 - 0.6)^2 + (-0.2 - 0.0)^2} \\
&= \sqrt{(-0.9)^2 + (-0.3)^2 + (-0.2)^2} = \sqrt{0.81 + 0.09 + 0.04} \\
&= \sqrt{0.94} \approx 0.9695
\end{aligned}$$

**Asymmetric Term:**

$$\begin{aligned}
u_{\text{asym}} - v_{\text{asym}} &= [-0.5 - 0.3, 0.0 - 0.5, 0.4 - 0.2] = [-0.8, -0.5, 0.2] \\
\text{ReLU}\left(\max(\cdot)\right) &= \text{ReLU}(0.2) = 0.2
\end{aligned}$$

**Final MRN Distance:**

$$d_{MRN}(u, v) = 0.9695 + 0.2 = \boxed{1.1695}$$

**Asymmetry Check $\left(d_{MRN}(v, u)\right)$** :

We now reverse the direction and compute $d_{MRN}(v, u)$.

$$\begin{aligned}
\|v_{\text{sym}} - u_{\text{sym}}\|_2 &= \|u_{\text{sym}} - v_{\text{sym}}\|_2 = 0.9695 \\
v_{\text{asym}} - u_{\text{asym}} &= [0.3 - (-0.5), 0.5 - 0.0, 0.2 - 0.4] = [0.8, 0.5, -0.2] \\
\text{ReLU}\left(\max(\cdot)\right) &= \text{ReLU}(0.8) = 0.8
\end{aligned}$$

$$d_{MRN}(v, u) = 0.9695 + 0.8 = \boxed{1.7695}$$

This example illustrates the inherent asymmetry captured by the MRN quasimetric:

$$d_{MRN}(u, v) = 1.1695 \quad \text{vs.} \quad d_{MRN}(v, u) = 1.7695$$

The directional gap arises from the residual term in the asymmetric subspace, which models the increased difficulty of transitioning from $v$ back to $u$ compared to the forward direction.

In summary, both IQE and MRN represent state-of-the-art approaches for learning quasimetric functions in reinforcement learning. IQE provides a highly structured and interval-based embedding with minimal parameterization, while MRN introduces a flexible residual-based construction that extends classical metric learning. Empirically, IQE tends to perform better in environments with structured navigation and short-to-medium horizon dependencies, whereas MRN is often more effective in handling long-horizon tasks and capturing fine-grained asymmetries.

## B  Impact of Positive and Negative Losses and the Distance Function

In this section, we analyze the individual and combined effects of the two core loss components used in training the quasimetric distance model: the **positive loss** $\mathcal{L}_{\text{pos}}$, which encourages small distance between successive states, and the **negative loss** $\mathcal{L}_{\text{neg}}$, which penalizes short distances to unrelated goal states. Our primary objective is to investigate whether training with only one of the two losses is sufficient or whether a *combination with a learnable balancing coefficient $\lambda$ is essential* for learning structured and effective distance functions in complex goal-conditioned tasks.

## B.1 Experimental Setup

We train three variants of the quasimetric model:

1. **QRL_pos:** Trained using only $\mathcal{L}_{\text{pos}}$ (positive supervision).

2. **QRL_neg:** Trained using only $\mathcal{L}_{\text{neg}}$ (negative supervision).

3. **QRL:** Trained using the full combined loss $\mathcal{L}_{\text{total}} = \mathcal{L}_{\text{neg}} + \lambda \cdot \text{stop\_grad}(\mathcal{L}_{\text{pos}})$, where $\lambda$ is learned dynamically during training.

Each model is evaluated on the AntMaze medium and large environments. After training, we plug the learned quasimetric into QuaD and evaluate the final goal-reaching success rate. We also monitor the evolution of average positive and negative distances over training to gain insight into learning dynamics.

Our results show a stark difference between the individual losses and the full model. When trained using only the positive loss term (`QRL_pos`), the quasimetric quickly collapses all distances to near zero, even for faraway or unreachable goals, resulting in the agent receiving misleadingly small guidance signals. Consequently, when this model is integrated into QuaD, it fails entirely to guide policy learning and yields **0% success** on all AntMaze environments.

On the other hand, training with only the negative loss (`QRL_neg`) causes the distances to inflate indiscriminately. The model lacks an anchor to define what *reachable* means, leading to overestimation of distances even for adjacent transitions. This again renders the model unusable for guiding goal-reaching behavior, and the agent also achieves **0% success** in all environments.

In contrast, the full QRL model—which combines both losses using an adaptive weighting parameter $\lambda$—learns to assign low distances to reachable states while maintaining high distances to unrelated goals. This dual structure provides the necessary contrast and balance for meaningful distance estimation. When this model is used in QuaD, it leads to **strong success rates**: over **59%** on `medium` mazes and over **31%** on `large` mazes. This highlights the importance of including both positive and negative supervision in training a quasimetric function that is robust, generalizable, and effective for guiding decision-making in sparse, goal-conditioned environments.

## B.2 Deeper Analysis: Why Both Losses Are Essential

The quasimetric learning formulation combines two distinct losses—**positive** and **negative**—each with a critical role in shaping the distance function $d(s, g)$. In this section, we provide an in-depth analysis of what these losses individually encode, why their combination is crucial, and what challenges arise when either component is removed.

### B.2.1 Positive Loss $\mathcal{L}_{\text{pos}}$: Grounding Through Reachable States

The positive loss anchors the distance function to reflect that transitions to nearby states (the next state) should have low distance. Formally, for a sampled transition $(s, a, s') \in \mathcal{D}$, we define:

$$\mathcal{L}_{\text{pos}} = \mathbb{E}_{(s,s')\sim\mathcal{D}} \left[ \left( \text{ReLU}(d_\theta(s, s') - 1) \right)^2 \right] \tag{15}$$

This loss penalizes the model when the estimated distance between $s$ and its immediate successor $s'$ is greater than a threshold (typically 1). Intuitively, this encourages that nearby and temporally consecutive states are pulled closer in the embedding space.

**However, this loss alone is insufficient.** If we remove the negative loss, the model quickly collapses to minimizing distances universally. In absence of any constraints on distant goals, the model trivially minimizes $\mathcal{L}_{\text{pos}}$ by mapping all inputs to the same embedding, thus flattening the distance landscape (as observed the distance estimation in figure 5:

$$d_\theta(s, g) \approx 0 \quad \forall s, g$$

Such a degenerate solution lacks any notion of long-range structure or directional effort. Empirically, this manifests as a zero success rate in planning tasks, since all states appear "close" and the model cannot distinguish between feasible and infeasible goals.

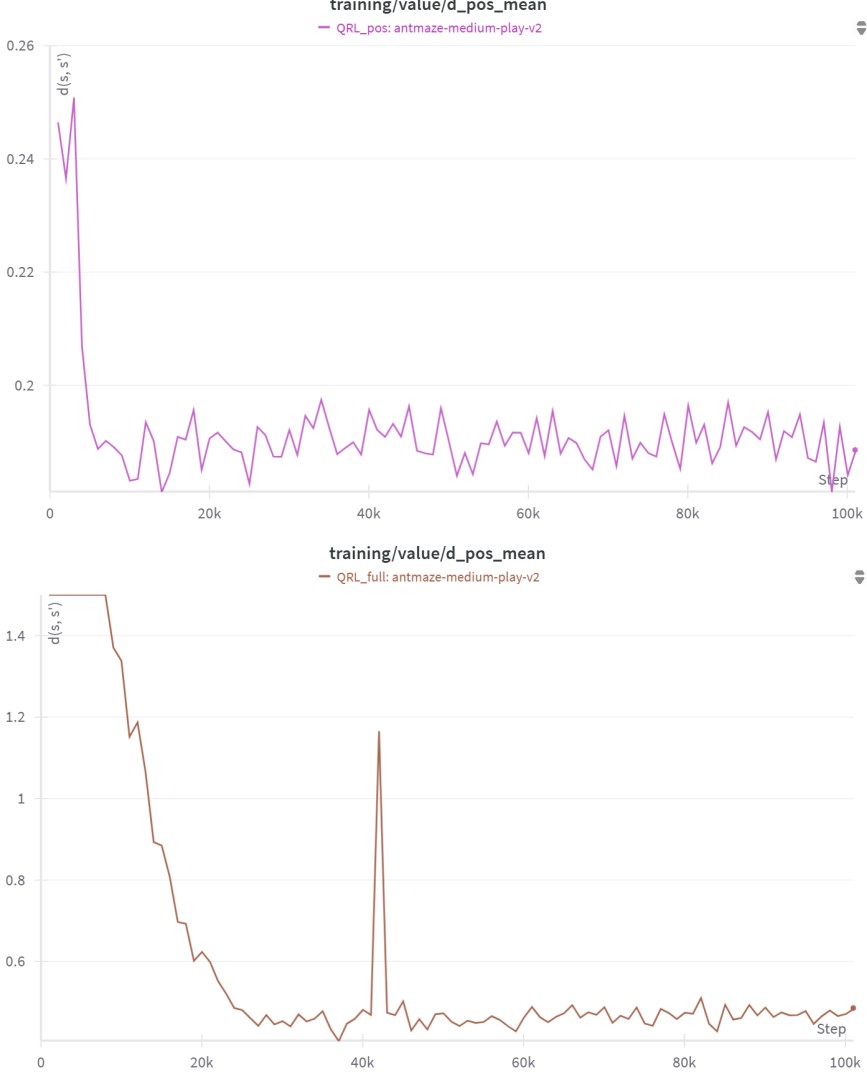

Figure 5: Comparison of the distance learned by both the models: QRL_pos(top) and QRL_full(bottom)

### B.2.2 Negative Loss $\mathcal{L}_{\text{neg}}$: Structural Discrimination from Unrelated Goals

To enforce global structure, we introduce a contrastive-like negative loss. Given a random goal $g \sim p_{\text{goal}}$ that is generally not reachable in one step from state $s$, we define:

$$\mathcal{L}_{\text{neg}} = \mathbb{E}_{(s,g)} \left[ 100 \cdot \text{softplus} \left( 5 - \frac{d_\theta(s,g)}{100} \right) \right] \tag{16}$$

This loss penalizes the model when it underestimates the distance to an unrelated goal. Its design has **Softplus** that smoothly penalizes low distances, avoiding hard margins or vanishing gradients.

**Yet again, this loss is insufficient on its own.** In the absence of $\mathcal{L}_{\text{pos}}$, the model has no lower bound or reference for what "nearby" looks like. The optimizer has no measure to maintain any local distance, leading to unbounded expansion of all distances:

$$d_\theta(s,g) >> 1$$

As shown in Figure 6 (top), this results in exploding distance estimates compared to QRL_full. Consequently, when used in a planning model like QuaD: it overestimates the distances for every goal, including feasible ones.

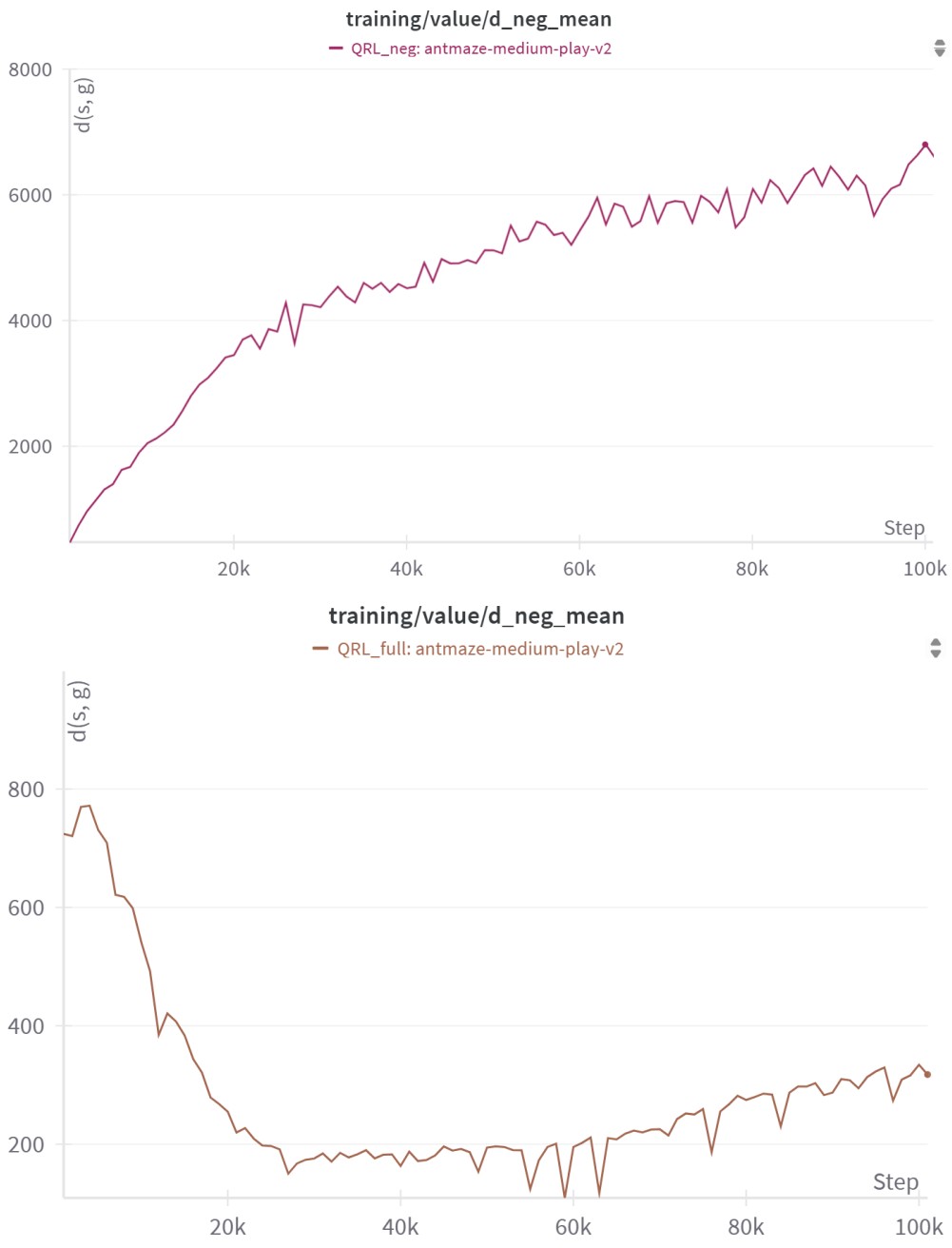

Figure 6: Comparison of the distance learned by both the models: QRL_neg(top) and QRL_full(bottom)

### B.2.3   Combined Loss with Learnable $\lambda$: Structured Grounding

Therefore, to maintain local distance as well as estimate global separation, we combine both losses using a learnable scalar $\lambda$, resulting in:

$$\mathcal{L}_{\text{total}} = \mathcal{L}_{\text{neg}} + \lambda \cdot \text{stop\_grad}(\mathcal{L}_{\text{pos}}) \tag{17}$$

This architecture enables the model to:

- Use **positive transitions** to *anchor* the distance scale near the agent.
- Use **negative goals** to *stretch* the embedding and introduce meaningful contrast.
- Adaptively **balance** both objectives based on current training dynamics.

The use of `stop_grad` ensures that $\lambda$ is updated to modulate the relative influence of the terms, without affecting the backward gradients through $\mathcal{L}_{\text{pos}}$.

Figure 5 and 6 clearly show that this formulation leads to a healthy separation of scales:

- $d_{\text{pos}}$ is small and stable: the model understands what reachable transitions look like.
- $d_{\text{neg}}$: the model learns to penalize unreachable or distant goals appropriately.

Together, they form a calibrated quasimetric landscape that captures both feasibility and directionality—critical for long-horizon sparse-reward tasks.

### B.3  Adaptive Weighting Behavior: Lambda Dynamics

In our full training setup, the weighting parameter $\lambda$ is learned alongside the quasimetric model. This adaptive scalar governs the trade-off between the positive loss $\mathcal{L}_{\text{pos}}$ and the negative loss $\mathcal{L}_{\text{neg}}$, enabling the model to dynamically prioritize either structure (via separation of random goals) or grounding (via continuity of reachable transitions) depending on the training stage.

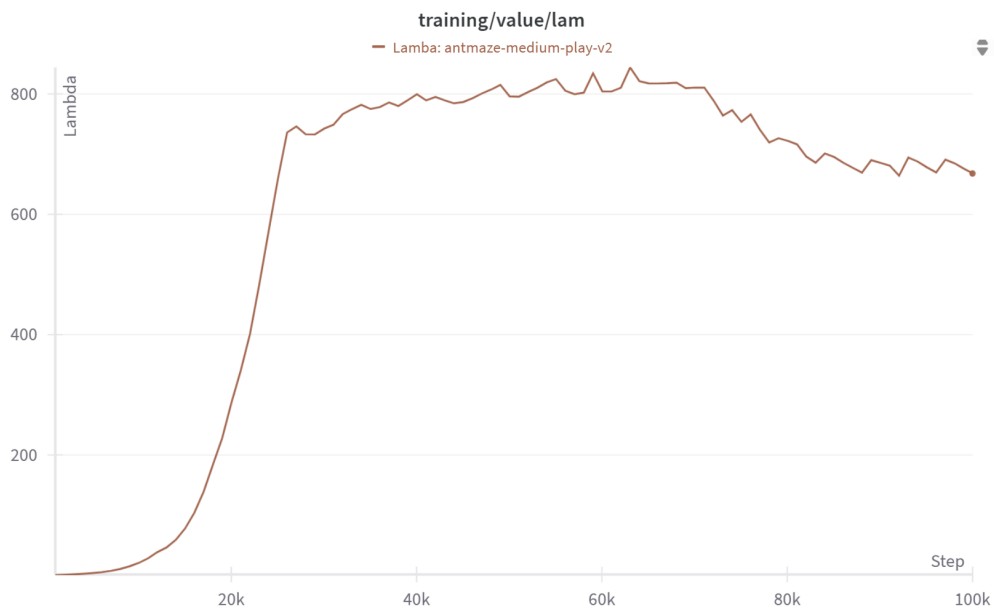

Figure 7: Learned value of $\lambda$ over training on `antmaze-medium-play-v2`.

Figure 7 shows how $\lambda$ evolves during training on the `antmaze-medium-play-v2` environment. In the early stages, we observe a sharp rise in $\lambda$, reaching a peak of approximately 800 by 30K steps. This indicates that the model initially prioritizes learning structure by emphasizing the negative loss term, pushing distances between unrelated start-goal pairs apart.

As training progresses, $\lambda$ begins to plateau and then gradually declines, suggesting a shift in focus toward the positive loss. This rebalancing is essential, and it prevents the model from arbitrarily inflating all distances due to the absence of grounding signals.

This dynamic behavior is not possible in fixed-weight baselines, where the relative importance of each loss remains static throughout training. By contrast, our learned-$\lambda$ strategy adapts to training dynamics, producing a more stable and robust quasimetric model.

### B.3.1 Summary

- Removing $\mathcal{L}_{\text{pos}}$: No sense of reachability or grounding. Everything is "far." Leads to overconservative agents.
- Removing $\mathcal{L}_{\text{neg}}$: No sense of difficulty or structure. Everything is "near." Leads to confused or overly optimistic agents.
- Combining both with $\lambda$: Learns directional distances that reflect true task difficulty.

Thus, the full QRL loss is *not just a sum of its parts*—it is a structured mechanism that provides both **calibration** and **contrast**. This makes it indispensable for learning meaningful quasimetric functions in complex environments.

## C Hyperparameters

### C.1 Quasimetric Network

We provide the hyperparameters used for training the Interval Quasimetric Embedding (IQE) and Metric Residual Network (MRN) value functions in Table 4. Most hyperparameters are set following standard configurations used in prior Quasimetric RL (QRL) (Wang & Isola, 2022a; Liu et al., 2023). Both IQE and MRN utilize a three-layer MLP with 512 hidden units per layer and layer normalization to ensure stable training. The latent dimension for both architectures is set to 512, with IQE using a per-component dimension of 8, which defines the number of interval embeddings used in the quasimetric representation. For the dual lambda loss, we set the margin parameter $\epsilon = 0.05$ across all environments.

Regarding dataset configurations, we set the probability of sampling random value goals to 1.0, ensuring diverse quasimetric learning, while future trajectory-based goal sampling is only applied for the actor policy. Additionally, geometric sampling is enabled for value function learning but is disabled for the actor function to avoid unintended bias in trajectory learning. The quasimetric loss formulation follows the original implementation in Quasimetric RL, where a softplus loss is used for negative distances, and a quadratic penalty is applied for positive distances exceeding 1.0. The full hyperparameter details are reported in Table 4.

| Hyperparameter | IQE (Interval Quasimetric Embeddings) | MRN (Metric Residual Network) |
|---|---|---|
| Learning Rate (lr) | $3 \times 10^{-4}$ | $3 \times 10^{-4}$ |
| Batch Size | 1024 | 1024 |
| Quasimetric Type | iqe | mrn |
| Value Hidden Dims | (512, 512, 512) | (512, 512, 512) |
| Latent Dimension | 512 | 512 |
| Dimension per Component | 8 | Not Applicable |
| Layer Normalization | True | True |
| Discount Factor | 0.99 | 0.99 |
| Epsilon for Lambda Loss | 0.05 | 0.05 |
| Quasimetric Function | Interval-Based IQE Metric | Residual Over Euclidean Distance |
| Distance Function | Mean and Max Aggregation | Euclidean + L-Infinity Metric |
| Alpha Parameter | Trainable via Sigmoid | Not Applicable |
| Distance Computation | Sorted Components with Negative Increments | Symmetric Euclidean + Asymmetric Max |

Table 4: Hyperparameter settings and architectural details for the two quasimetric network versions: IQE and MRN.

### C.2 Quasimetric Decision Transformer

We summarize the hyperparameters and architectural details of the Quasimetric Decision Transformer (QuaD) in Table 5. Most hyperparameters align with standard Decision Transformer (DT) (Chen et al., 2021) configurations. The model is trained using a sequence length of 20 with a transformer-based architecture consisting of 4 causal attention blocks, each using 8 self-attention heads and a dropout probability of 0.1. The

embedding dimension is set to 128, with separate state-goal, action, and quasimetric distance embeddings to enhance representation learning.

For quasimetric learning, we experiment with two quasimetric functions: Interval Quasimetric Embeddings (IQE) and Metric Residual Networks (MRN), where the latent dimension is set to 512. The quasimetric-guided actor policy is trained using DDPG+BC by default, but we also evaluate Advantage-Weighted Regression (AWR) loss settings in ablation studies. The quasimetric prediction model is integrated into the autoregressive transformer framework, where quasimetric distances are computed at each timestep and embedded into the transformer sequence model.

Regarding training settings, we follow standard DT training configurations, using an Adam optimizer with a learning rate of $8 \times 10^{-4}$, weight decay of $1 \times 10^{-4}$, and gradient clipping at 0.25 to ensure stable training. The quasimetric target values replace the standard return-to-go (RTG) formulation, providing a structured goal-reaching metric for improved sequence modeling. We evaluate the QuaD framework across six AntMaze environments from D4RL. The full list of model-specific and training-specific hyperparameters is presented in Table 5.

| Hyperparameter | Value |
|---|---|
| **General Training Settings** | |
| Batch Size | 64 |
| Training Steps | 100,000 |
| Evaluation Episodes | 100 |
| Episode Length | 1,000 |
| Evaluation Interval | 10000 |
| Discount Factor ($\gamma$) | 0.99 |
| Learning Rate | $8 \times 10^{-4}$ |
| Weight Decay | $1 \times 10^{-4}$ |
| Adam Beta Parameters | (0.9, 0.999) |
| Gradient Clipping | 0.25 |
| Warmup Steps | 10,000 |
| **Decision Transformer Model** | |
| Sequence Length | 20 |
| Number of Transformer Blocks | 4 |
| Hidden Dimension ($h_{\mathrm{dim}}$) | 128 |
| Number of Attention Heads | 8 |
| Dropout Probability | 0.1 |
| Attention Heads | 8 |
| **Quasimetric Network** | |
| Quasimetric Type | IQE / MRN |
| Latent Dimension | 512 |
| Actor Loss Type | AWR / DDPG+BC |
| Alpha Scaling Factor | 0.003 |
| Constant Standard Deviation | True |

Table 5: Hyperparameter and Architectural Details of the Quasimetric Decision Transformer (Quad).

