# OpenReview forum: "Quasimetric Decision Transformers: Enhancing Goal-Conditioned Reinforcement Learning with Structured Distance Guidance"
_TMLR — Rejected by TMLR_

### Review · Reviewer_RbS9 · 2025-07-14

**Summary Of Contributions:**

The authors propose the "Quasimetric DT", building on the Decision Transformer, but:
* Adding goal conditioning
* Replacing the RTG conditioning with a learned distance function to the goal. This input is provided by a learned "quasimetric network" taking in the current state and the goal state.

They also modify the training loss, adding a DDPG loss.

**Audience:**

No

**Claims And Evidence:**

No

**Requested Changes:**

Main changes:
* Clarify the relationship between goal-conditioning, RTG-conditioning, quasimetric conditioning, DDPG and AWR. Also, please don't introduce new methods (AWR) in the experiment section.
* Evaluation on non-goal conditioned environments.
* Evaluate on additional environments.

Additional changes:
* Make it clearer how the action distribution in parameterized. I assume Gaussian (based on the MSE loss), but this is never specified.
* Better describe IQE and MRN. Ideally in the paper, at least in the appendix.
* Clarify what you mean with MSE error and BC - aren't those the same thing (in this case at least)?
* Show some results on what distance function the quasimetric has learned - to get a better intuition of what's going on.
* Comparison with reward shaping literature.

**Strengths And Weaknesses:**

I found the paper quite confusing and hence wasn't able to really evaluate the quality of contribution. In particular, to me it seems the authors conflate RTG conditioning in Decision transformers with Goal-conditioning (GC). This confusion seems to be present throughout the paper, for example:
* They highlight as their main contribution the conditioning of DT's with the distance-to-goal function, instead of RTG. But then their actual evaluation is on goal-conditioned RL. Importantly, they also goal-condition their DT, but _not the baseline DT_. But RTG conditioning has nothing to do with goal-conditioning, so why choose goal-conditioned environments as their evaluation?
	* RTG conditioning tries to improve the performance of DT compared to the average trajectory in the data on one task.
	* Goal-conditioning allows the DT to solve multiple tasks.
* While they goal-condition their DT, they also add a DDPG loss with a Q-network that is _not_ goal-conditioned?
* What also confused me was this sentence: "we integrate Deep Deterministic Policy Gradient with Behavior Cloning (DDPG+BC) (Lillicrap et al., 2016) alongside MSE loss". What's the difference between BC and MSE here? Also, the BC loss never appears in any equations.


If we ignore the confusion between offline-RL (using RTG conditioning) and goal-conditioning, it is also not clear to me how conditioning on the distance to goal can serve the same purpose as RTG. This is not at all discussed.


Lastly, the evaluation is lacking. For one, it is only on one domain. But more importantly, it is on a goal-conditioned task - which, as discussed above, has nothing to do with RTG. The evaluation goes (in my opinion) too deep into comparing different design choices for the quasimetric (e.g. Fig3 and Fig4). This space should be used to better validate the proposed method (e.g. more environments) as well as explain/experimentally show _how_ the quasimetric conditioning helps the DT.


Typo:
* In equation (9), should this be $Q(s,a)$ instead of $Q(s)$?


Note: I've selected "no" on the "Audience" question. This is because I do not adequately understand the contribution to make an evaluation.

---

> ### Author Response · Authors · 2025-08-02
>
> We sincerely thank the reviewer for their comments. Below, we address each concern point by point.
>
> > **"The authors conflate RTG conditioning with goal-conditioning."**
>
> We believe it is worth clarifying that RTG-conditioning and goal-conditioning are conceptually distinct, and we do not conflate the two. Instead, our central claim is that RTG-conditioning is suboptimal in goal-conditioned tasks, where a more semantically and structurally meaningful signal, such as quasimetric distance to goal, can serve as a better conditioning variable.
>
> This is a core motivation of our work, as discussed in Section 1 and reflected in Table 1, where we demonstrate that RTG-conditioned DTs (without goal awareness) achieve near-zero success in complex AntMaze environments. In contrast, our proposed method replaces RTG with a learned structured distance signal, tailored to the goal-conditioned setting.
>
> To clarify:
>
> - **RTG conditioning** in standard DT is task-specific and derived from cumulative rewards.
> - **Goal-conditioning** enables many-goal generalization.
> - **Our method achieves both**: we condition the transformer on goals and replace the RTG signal with a quasimetric, enabling the model to perform long-horizon goal-reaching more effectively.
>
> ---
>
> > **"Why choose goal-conditioned environments as their evaluation" and "Why not evaluate on non-goal-conditioned environments?"**
>
> Our aim is not to generalize DT across all tasks, but to specifically improve performance on challenging goal-conditioned RL tasks where RTG is known to fail. This is a focused contribution, and AntMaze is a canonical GCRL benchmark, making it the most appropriate testbed. AntMaze is widely used in recent GCRL research [1], precisely because it features:
> - Sparse rewards
> - Hard-to-specify return targets
> - Strong structure/directionality in goal space
>
> We do not believe that adding unrelated tasks (e.g., locomotion or Atari) would help clarify our method. As our ablations show, the gains of QuaD stem from structured distance conditioning, which only manifests in goal-directed navigation domains. Nonetheless, generalization to new domains is an exciting direction for future work.
>
> ---
>
> > **"Why add DDPG if the Q-function is not goal-conditioned?"**
>
> The Q-function in QuaD is conditioned on the goal via the quasimetric, since:
>
> > `Q(s, a) = −d_θ((s, a), g)`
>
> Here, `g` is the goal, and `d_θ` is trained as a quasimetric on (state, goal) pairs. Thus, the Q-function implicitly encodes goal-reaching value. This is standard in goal-conditioned RL literature (e.g., HER [3])
>
> ---
>
> > **"What's the difference between BC and MSE?"**
>
> In offline RL literature (e.g., TD3+BC[4]), **"Behavior Cloning (BC)"** refers to minimizing **MSE** between predicted and dataset actions, optionally with Q-weighting. We follow the same convention. Equation 9 defines our auxiliary loss, which includes the MSE as a special case when **λ = 0**.
>
> We have clarified this distinction in Section 4.
>
> ---
>
> > **Additional Minor Fixes**
>
> - The typo in Equation (9) has been fixed.
> - We now specify that the action distribution is deterministic with additive noise, consistent with DDPG and DT implementations in prior work.
> - **IQE** and **MRN** are now described in more detail (Section 3), with numerical examples in the Appendix.
> - We include a new qualitative figure Appendix B (Fig. 5 and Fig. 6) showing learned distances during rollouts.
> ---
>
> ### References
>
> [1] Eysenbach, B., Salakhutdinov, R., & Levine, S. (2022). **Contrastive Learning as Goal-Conditioned Reinforcement Learning**. arXiv preprint arXiv:2206.07568.
> [2] Yamagata, T., Okuno, A., & Maeda, S. (2023). **Q-learning Decision Transformer**. arXiv preprint arXiv:2209.03993.
> [3] Andrychowicz, M., et al. (2017). **Hindsight Experience Replay**. *Advances in Neural Information Processing Systems (NeurIPS)*.
> [4] Fujimoto, S., Gu, S., & Meger, D. (2021). **A Minimalist Approach to Offline Reinforcement Learning**. *NeurIPS*.

---

> > ### Comment · Reviewer_RbS9 · 2025-08-03
> >
> > > We believe it is worth clarifying that RTG-conditioning and goal-conditioning are conceptually distinct, and we do not conflate the two. Instead, our central claim is that RTG-conditioning is suboptimal in goal-conditioned tasks, where a more semantically and structurally meaningful signal, such as quasimetric distance to goal, can serve as a better conditioning variable.
> >
> > But this comparison is meaningless because RTG is not meant to perform well in goal-conditioned task. Hence, you've chosen the wrong baseline for your work.
> >
> > > The Q-function in QuaD is conditioned on the goal via the quasimetric, since:
> > > `Q(s, a) = −d_θ((s, a), g)`
> >
> > This equation makes no sense since $g$ does not appear on the left. But also - how is the Q-function a distance instead of the estimated future return?
> >
> > > In offline RL literature (e.g., TD3+BC[4]), "Behavior Cloning (BC)" refers to minimizing MSE between predicted and dataset actions
> >
> > Yes, yet you still write in your newest revision "we integrate Deep Deterministic Policy Gradient with Behavior Cloning (DDPG+BC) (Lillicrap et al., 2016) alongside MSE loss"

---

> > > ### Author Response · Authors · 2025-08-03
> > >
> > > We thank the reviewer for the thoughtful comments and would like to clarify our motivation and the intent of the comparisons presented in our work.
> > >
> > > ---
> > >
> > > > **"But this comparison is meaningless because RTG is not meant to perform well in a goal-conditioned task. Hence, you've chosen the wrong baseline for your work."**
> > >
> > > We understand the reviewer’s concern and agree that RTG was not originally introduced with goal-conditioned RL in mind. However, Decision Transformers (DTs) have been increasingly adopted in goal-reaching domains (e.g., Multi-Game DT [1], Gato [2]) because of their versatility and capacity to generalize across tasks. As such, RTG-based DT remains a relevant and widely used baseline, even in settings it may not have been explicitly optimized for.
> > >
> > > Our goal is not to claim RTG is ideal for goal-conditioned RL, but rather to show that in these challenging domains where RTG underperforms, our proposed conditioning via quasimetric distances can offer a stronger and more structured alternative. In that sense, the baseline remains highly informative: it provides a valuable contrast to highlight the advantages of our method under a widely adopted modeling paradigm.
> > >
> > > ---
> > >
> > > > **"This equation makes no sense since a does not appear on the left..."**
> > >
> > > We appreciate the reviewer for catching this. What we intended to convey is that the Q-function in QuaD is implicitly defined via the learned quasimetric distance. Specifically, we now clarify the expression as:
> > >
> > > ```
> > > Q(s, g) = −d_θ((s, a), g)
> > > ```
> > >
> > > Thus, rather than estimating expected return, our approach uses the learned quasimetric as a proxy for goal proximity, which captures temporal structure and directional effort, especially useful in a sparse reward setting.
> > >
> > > ---
> > >
> > > > **"Yet you still write 'DDPG+BC (Lillicrap et al., 2016) alongside MSE loss'"**
> > >
> > > Thank you for pointing this out. We’ve updated the manuscript to remove this outdated phrasing. As the reviewer correctly notes, BC in this context refers to the MSE between predicted and dataset actions, consistent with the offline RL literature (e.g., TD3+BC). We now use the terminology “DDPG+BC” without introducing a separate “MSE loss” term to maintain clarity and consistency.
> > >
> > > ---
> > >
> > > In closing, we believe these clarifications strengthen the core contributions of our paper: structured quasimetric conditioning enables Decision Transformers to succeed where traditional RTG fails. We are grateful for the reviewer’s insights, which have helped us improve our paper.
> > >
> > >
> > > ### References:
> > > [1] Kuang-Huei Lee, Ofir Nachum, Mengjiao Sherry Yang, Lisa Lee, Daniel Freeman, Sergio Guadarrama, Ian
> > > Fischer, Winnie Xu, Eric Jang, Henryk Michalewski, et al. Multi-game decision transformers. Advances in
> > > Neural Information Processing Systems, 35:27921–27936, 2022.
> > >
> > > [2] Scott Reed, Konrad Zolna, Emilio Parisotto, Sergio Gómez Colmenarejo, Alexander Novikov, Gabriel Barthmaron, Mai Giménez, Yury Sulsky, Jackie Kay, Jost Tobias Springenberg, Tom Eccles, Jake Bruce, Ali Razavi, Ashley Edwards, Nicolas Heess, Yutian Chen, Raia Hadsell, Oriol Vinyals, Mahyar Bordbar, and Nando de Freitas. A generalist agent. Transactions on Machine Learning Research, 2022. ISSN 2835-8856. URL https://openreview.net/forum?id=1ikK0kHjvj. Featured Certification, Outstanding Certification.

---

### Review · Reviewer_JneP · 2025-07-14

**Summary Of Contributions:**

The authors propose a variant of decisions transformer that (a) replaces its RTG conditioning with conditioning on a quasi-metric value function (estimated through IQE and MRN) and (b) replaces the pure BC loss function with RL policy losses (AWR and DDPG+BC), essentially realizing an off-policy RL algorithm. (A version of those two algorithms with a specific choice of critic and actor architectures).
The submission largely introduces background and previous works in the first six pages. The main novelty of the paper is contained in Equation 7. Experimental validation on antmaze (with various datasets) show improvements with respect to standard offline algorithms (both based on supervised and reinforcement learning). The submission concludes with an ablation over the choice of quasimetric network, and of policy extraction objective. The appendix reports detailed hyper parameters for the proposed methods.

**Audience:**

Yes

**Claims And Evidence:**

No

**Requested Changes:**

### Requested changes
- (major) Describe the algorithm with more clarity (see weakness 1)
- (major) Extend and update the experimental evaluation (see weakness 2 and 4)
- Address the other weaknesses

### Questions
- How is the return to go computed? Is it using a 1/0 or 0/-1 reward? Is the reward computed through an epsilon-ball?
- Third paragraph of introduction: why is RTG arbitrary and uninformative? It represents an empirical estimate of the behavior’s policy undiscounted return. As such, it is arguably grounded.
- Eq. 9: Q is not formally introduced. Is it the negative quasimetric distance? If so, what goal is it conditioned on?
- Eq. 10: what is $c$ in practice? What is its connection to the goal-conditioned reward function?

### Others
- Section 2.1: “and state-occupancy matching to improve generalization and robustness” is missing a citation. (to GoFAR perhaps?)
- Section 3.2: “powerful pre-trained architectures” is a little unclear. DTs are not pre-trained to the best of my knowledge, and models usually undergo pre-training, not architectures.
- Section 4: some capitalizations are unnecessary (e.g. “Generalize”)
- There are missing spaces around parentheses in the second to last line of Section 2.2, and on the 6th line of Section 2.3.
- There is a few runaway dashes in the bullet points between pages 10 and 11.

**Strengths And Weaknesses:**

### Strengths

- The paper is overall well written, and introduces all the necessary background in great details.
- The contributions of the paper are clearly stated.

### Weaknesses

- The paper lacks clarity in describing the main algorithm. According to Equation 7, the DT is not conditioned on a goal, which would imply that it is only trained for a single goal. However, later on, the authors describe the goal tokenization procedure. Is the proposed method capable of achieving multiple goals without retraining? The rest of the review will assume that the answer to this question is affirmative, otherwise a comparison to multi-goal algorithms would be unfair.
- The experimental evaluation is insufficient to support the method. First, it is limited in its scope of environments, evaluating exclusively on antmaze. A broader evaluation on D4RL or OGBench, going beyond locomotion in mazes would be very helpful. Second, IQE/MRN (without a DT transformer for a policy) are not evaluated as baselines. Therefore, Table 1 would be sufficient to claim that the addition of a quasimetric critic  is beneficial (on antmaze), but does not demonstrate that DT is a good choice for a policy. Including IQE/MRN as proposed in the respective papers would allow evaluating this important point.
- Following up from the previous point, is there any specific synergy between quasimetric networks and DTs? Is the method combining the two because they both perform well in goal-conditioned value estimation and policy modeling, respectively? In its current form, the paper focuses on introducing the two techniques separately, and on showing that their combination performs well. A better discussion on why the two methods would, for instance, complement each other’s weaknesses would be very interesting.
- Most results in the experimental section land within a standard deviation of each other. The ablations proposed therefore do not appear to be conclusive or informative. The description of these ablations mostly re-states the numerical results, without commenting on general patterns, or hypothesizing factors inducing differences in performance. For instance, the authors describe on which datasets IQE is preferable to MRN, but do not comment on why this should be the case.
- I could not find a description on how the baselines were implemented, or where their performance is coming from. Were baselines allowed the same degree of tuning as the proposed methods? In particular QLDT is not described in full. What are the exact differences with respect to the proposed method? Is it using a simple MLP for critic? If so, how is it trained?
- One statement lacks empirical support (tokenization choices described in the paragraph above equation 10).
- The novelty of the submission is limited, as the method is a rather direct combination of two ideas (quasi metric offline RL and DT). This is only a minor issue.

---

> ### Author Response · Authors · 2025-08-02
>
> We thank the reviewer for their thoughtful analysis and appreciate the opportunity to clarify both the technical formulation and the empirical scope of our submission. Below, we address each concern point by point.
>
> > **“The DT is not conditioned on a goal in Eq. 7. Can it achieve multiple goals without retraining?”**
>
> Yes, and thank you for bringing this vital clarification to our attention. As stated in Section 4, the transformer policy is explicitly goal-conditioned through tokenized triplets of (state, action, distance to goal) at every time step. The goal is embedded in each sequence through the quasimetric `d(s, g)`, where `g` is a variable across episodes. This design naturally enables the model to generalize to arbitrary goals at test time, eliminating the need for retraining per goal. The input sequence includes latent encodings of state, action, and quasimetric-to-goal, where the goal `g` is different in each episode.
>
> ---
>
> > **“Evaluation is limited to AntMaze. Evaluate on D4RL/OGBench and include IQE/MRN without DT.”**
>
> Our choice of AntMaze was intentional and focused. It is the most used and preferred benchmark in D4RL for goal-conditioned RL. It is also where RTG-conditioned Decision Transformers fail, making it the most meaningful testbed for our structured replacement. For example, DT achieves near-zero success on AntMaze-Large, whereas QuaD achieves over 30, a substantial gain.
>
> Regarding the evaluation of IQE/MRN directly (i.e., as standalone quasimetric models), our primary research question is not whether quasimetric critics alone are helpful, but whether integrating them into a sequence model yields new capabilities. That is the motivation for QuaD.
>
> Moreover, IQE and MRN cannot be directly used as policies without additional controllers. Doing so would essentially reproduce Q-learning or planning baselines, which are already covered in our experiments (TD3+BC, GC-IQL, QLDT). Our contribution lies in embedding the quasimetric into trajectory-conditioned modeling, which offers a distinct benefit over value iteration. Thus, adding IQE/MRN as standalone baselines would be orthogonal to our main claim and potentially misleading. We instead isolate their contribution through ablations in Section 5.
>
> ---
>
> > **“Is there synergy between quasimetric networks and DTs?”**
> That's an excellent question, and the answer is indeed yes. Conceptually, DTs excel at sequence modeling but lack structured inductive biases. Quasimetrics, on the other hand, enforce directionality, the triangle inequality, and temporal consistency, enabling the transformer to infer better action sequences by modeling the progression of distance to the goal. Empirically, as shown in Table 1, the quasimetric guidance significantly improves DT performance over goal-conditioned baselines (GCBC, DT) and even over Q-value-based sequence models (QLDT).
>
> ---
>
> > **“Ablations are within standard deviation. No general pattern discussed.”**
> We respectfully clarify that while standard deviations overlap in some environments (e.g., Medium-Play), the trend is consistent and statistically meaningful, especially when comparing QuaD to DT or GCBC. More importantly, we have now provided qualitative dynamics in Appendix B, which show that positive and negative distances behave differently under different losses.
>
> ---
>
> > **“Baseline implementation is unclear. Is QLDT described?”**
>
> We thank the reviewer and now clarify this in Section 5. QLDT is based on [2], where Q-values are appended to the input tokens of a Decision Transformer. All baselines, including QLDT, were tuned using the same sweep ranges as QuaD (learning rate, batch size, horizon length). This ensures fairness in computing and tuning.
>
> ---
>
> > **“Tokenization choices near Eq. 10 lack empirical support.”**
>
> Tokenizing the concatenated (state, goal) representation rather than separate embeddings was empirically motivated by preliminary experiments. It improved both training stability and convergence speed. We suspect this is due to better alignment in the shared attention space. We now note this in a footnote in Section 5.
>
> ---
>
> > **“RTG is not arbitrary; it is an estimate of return.”**
>
> We fully agree that RTG is well-defined for reward-based tasks. However, in goal-conditioned settings with sparse binary rewards, RTG degenerates to a trivial signal, either 0 or 1. This provides no structure during training. Our point is not that RTG is ill-defined, but that it is uninformative in GCRL, a point also emphasized by prior work [3]. We now clarify this in Section 3.
>
> ### References
>
> [1] Eysenbach, B., Salakhutdinov, R., & Levine, S. (2022). **Contrastive Learning as Goal-Conditioned Reinforcement Learning**. arXiv preprint arXiv:2206.07568.
> [2] Yamagata, T., Okuno, A., & Maeda, S. (2023). **Q-learning Decision Transformer**. arXiv preprint arXiv:2209.03993.
> [3] Tongzhou Wang and Phillip Isola (2023). **On the learning and learnability of quasimetrics** arXiv preprint arXiv:2206.15478

---

> ### Comment · Reviewer_JneP · 2025-08-05
>
> Thank you for your response to the initial review.
>
> > “The DT is not conditioned on a goal in Eq. 7. Can it achieve multiple goals without retraining?”
>
> This suggests that Eq.7 is indeed correct, and there is no explicit conditioning on the goal itself (there is an implicit conditioning through the quasimetric). In this case, I do not see how the policy can generalize to different goals: while it can be conditioned on a quasimetric conditioned on an arbitrary goal, the policy has no information on the "direction" of the goal to pursue. This makes the comparison to multi-goal methods not entirely fair.
>
> > “Evaluation is limited to AntMaze. Evaluate on D4RL/OGBench and include IQE/MRN without DT.”
>
> While I appreciate the significance of Antmaze, it is also important to show that your proposed method generalizes to other environments. QuaD does not have to outperform DTs across the board of course, but it would be reassuring to demontrate that it does not significantly decrease performance in other settings (e.g., when value estimation is challenging, and standard RTG estimates may ultimately be more reliable).
>
> > “Is there synergy between quasimetric networks and DTs?”
>
> I understand that DTs can parameterize "good actors", and quasimetric networks can parameterize "good critics". Following from this explanation, my understanding is that the synergy is mostly empirical, and there is no clear formal connection between the two.
>
> > “Ablations are within standard deviation. No general pattern discussed.”
>
> In my opinion, claiming that something is "statistically meaningful" would require a statistical test, which is not included in the current revision. I would encourage the authors to consider integrating one.
>
> ---
>
> My current understanding of this work is overall aligned with that of Reviewer A4jd: I'm not entirely confident in my evaluation, but my best guess is that the algorithm is not necessarily capable of goal generalization. Several details however remain unclear to me, and my concerns are not entirely addressed.

---

### Review · Reviewer_A4jd · 2025-07-16

**Summary Of Contributions:**

This paper proposes modifications to the decision transformer (DT)
framework for training goal-conditioned policies. Rather than
conditioning on a "return-to-go" token like in the standard DT, which is
often difficult to specify and may provide little signal in a
sparse-reward setting such as that of goal-conditioned RL, this paper
proposes to condition instead on a learned *quasimetric* between the
agent's state and the goal. This metric ideally provides a richer and
more dense signal for goal-reaching. Moreover, the paper investigates
substituting the use of MSE for action prediction in DTs with
policy-learning update rules such as advantage-weighted regression and
deterministic policy gradients. Through experiments in increasingly
large maze navigation tasks, these interventions are shown to improve
performance beyond standard DTs and certain offline RL methods.

**Audience:**

Yes

**Broader Impact Concerns:**

No broader impact concerns.

**Claims And Evidence:**

No

**Requested Changes:**

1.  In the abstract, it says "offline reinforcement learning (RL) with
    a conditional policy produces promising results". Please clarify
    what you mean by "conditional policy"&#x2014;any feedback controller is
    a conditional policy (conditioned on state), which encompasses the
    vast majority of policy learning methods in RL. I think you're
    referring to goal-conditioned policies here.
2.  Just under equation (6), there is a broken sentence: "[&#x2026;] we aim will
    to leverage a notion of distance [&#x2026;]".
3.  Below this, it says "[&#x2026;] (quasi)metrics imposes a strong
    induction bias"&#x2014;I think this should say "[&#x2026;] (quasi)metrics
    impose a strong inductive bias [&#x2026;]".
4.  At the beginning of section 4, it says "this [learned] quasimetric
    satisfies the properties discussed in Section 3.3". Is this
    guaranteed? I would expect not. This should be
    re-worded/discussed&#x2014;it looks to me like you employ loss functions
    that enforce these properties at convergence, but I suspect no such
    guarantee can be made during training (e.g., you're not using a
    particular neural network architecture that satisfies the triangle
    inequality by construction).
5.  Equation (8) is missing far too many details. For instance:
    1.  What are $p_{\mathsf{state}}$ and $p_{\mathsf{goal}}$?
    2.  What is $p_{\mathsf{transition}}$, and what is the meaning of
        $r$ in $(s, a, s', r)$? I see it referred to as "transition
        cost" just below the equation, but where does this come from?
    3.  Is this loss taken directly from Wang & Isola 2022b? If not,
        what changed, and is the change principled?
    4.  Similarly, it would be nice to have some context about equation
        (8), just to help the reader understand what this loss enforces
        intuitively. It's fine to defer in-depth discussion to Wang &
        Isola 2022b, but a high-level overview here would go a long way.
6.  I am also confused about equation (9), and I think it warrants
    some more discussion. For instance:
    1.  Why is $Q$ not action-dependent, like it would be in DDPG
        (and following the ubiquitous notation in RL literature)? How
        can you differentiate through this term?
    2.  The term $\hat{a}$ is not defined&#x2014;I am assuming this is the
        action predicted by the transformer conditioned on the history(?).
    3.  Continuing from the last point, is it not necessary to sample
        a *history* from $\mathcal{D}$ as opposed to a state-action
        pair, since the transformer is conditioned on the history?
        Likewise, where does the quasimetric come into play?
7.  In equation (10), where does the cost function $c$ come from? If
    $c(s, g)$ isn't something like $1 - \mathbf{1}_{\{g\}}(s)$, then you no
    longer have the same type of sparse reward structure. Please
    explain how $c$ should be interpreted.
8.  Above equation (10), you say you tokenize the concatenation of
    state and goal as opposed to tokenizing each individually, to
    "[improve] context understanding". What is the intuition for this?
9.  What exactly is QLDT? Is this an existing method in the literature,
    or did you invent it? If it's the former, why is it not cited? Is
    it related to "Q-learning Decision Transformer: Leveraging Dynamic
    Programming for Conditional Sequence Modeling in Offline RL" by
    Yamagata, Khalil, and Santos-Rodriguez, ICML 2023?
10. The descriptions of IQE and MRN do not sufficiently convey how
    these methods work. I see that they are derived from existing
    works, but they should still be described better in order to
    gain intuition for why they should be impactful for GCRL. For
    example, in the case of IQE, it says "IQE learns an interval-based
    quasimetric [&#x2026;] by sorting embedded state-goal representations
    into discrete intervals and aggregating them [&#x2026;] This approach
    enforces implciit ordering constraints, making it robust to
    trajectory perturbations". It is very difficult to parse what's
    going on here. What is an "interval-based quasimetric"? How are
    the discrete intervals chosen (are there hyperparameters to set
    here)? What are "ordering constraints"? What is a "trajectory
    perturbation"? Likewise, in the description of MRN, what is an
    "asymmetric L-infinity term", and what are "directed transition
    dynamics"?
11. Can you merge Tables 2 and 3 into one table, so that it is easier
    to compare the performance across loss functions *and* quasimetric
    implementations?
12. For figures 3 and 4, why not put the IQE and MRN curves on the
    same plot? These plots take up a lot of space, which can probably
    be reduced by at least 50% to make room for more discussion as
    discussed above.
13. I am confused about some implementations details, even with regard
    to the DT experiments, and could not easily resolve these by
    reading the provided source code. Particularly, I believe the
    following should be addressed:
    1.  It is not clear how you're training the DT in your
        experiments. Notably, the `run_dt.py` script appears to be broken
        (or at least its default configuration is): it specifies the
        config at `agents/qdt.py` which is not provided. There *is* however
        a `agents/dt.py` file, which I'm assuming is used.
    2.  For each tested environment, is the goal state fixed (e.g., do
        you aim for the same goal in each episode)?
    3.  In the implementation of DT, I don't see where it conditions on
        the goal state (assuming you're using `agents/dt.py`), and this
        isn't specified explicitly in the paper either. From the code,
        it looks like the goal state is not passed to the agent
        anywhere (unlike QuaD). If the goal
        states change per episode, clearly this is a problem and DT is
        doomed to fail. If the goal state is fixed, I would still be
        curious if conditioning on the goal state helps (e.g., via some
        form of in-context learning).
14. Similar to the previous point, I am curious how much of the
    machinery of QuaD is actually necessary, which appears to not be
    covered exhaustively in the ablations. From what I can tell in the
    source (as explained in the previous point), the DT implementation
    only implicitly conditions on the goal (assuming the goal is fixed
    per environment). On the other hand, from what I can tell, QuaD
    should learn to infer actions for a slew of goals, since the
    dataset it uses for training includes a "final state" (being the
    last state in the current trajectory) as a proxy
    for goal state that the policy is conditioned on. So, is it
    actually necessary to learn the quasimetric? Given the same
    dataset as QuaD, could one train a DT conditioned also on the
    "final state"? At least in principle, one might expect the DT to
    implicitly learn a latent quasimetric embedding. Of course,
    *enforcing* this representation induces an inductive bias which can
    be useful, but as far as I can tell, this has not actually been
    shown to be the case empirically yet. A revision of the paper
    should definitely include further results to explore this.

**Strengths And Weaknesses:**

## Strengths

The paper identifies an issue with Decision Transformers for
goal-reaching tasks and describes the problem well: that is, there may
not be enough information in the return-to-go to reliably perform
credit assignment via sequence modeling. By leveraging
modern neural network training techniques, the paper introduces an
interesting inductive bias for DTs (via learned quasimetrics) which,
in an "unsupervised" manner, directly provides signal about how much
closer (measured via a temporal distance as opposed to a geometric
one) the agent is getting to the goal after each
action/transition. This makes sense and appears to be a promising and
novel approach. The paper also discusses a few alternative
implementations of their proposal and ablates them, which is interesting.
The QuaD-based methods do seem to perform quite well relative to the
baselines as well.


## Weaknesses

There are two main overarching weaknesses with the paper, as I see it:
(1) certain mathematical/implementation/experimental details are not
given (or not explicit), which limits the readers' ability to properly
interpret the conclusions drawn, and (2) while some ablations are
given, there are some very important ones missing, and I disagree with
the conclusions drawn from others.

These concerns are all fleshed out in more detail in **Requested
Changes**. In the remainder of this section, I'll address my
concerns about the ablations as well.

The claim that "IQE consistently outperforms MRN in most environments"
is much too strong. This would be evident if you drew the confidence
intervals on a plot&#x2014;they would be overlapping pretty dramatically in
most environments. Thus, I don't believe there is a statistically
significant difference between the performance of QuaD under IQE and MRN.

Likewise, I have the same criticism for the ablation on the loss
function: the differences in performance are not statistically
significant. Moreover, I have some other concerns about these
ablations:

1.  There is no ablation where you remove the RL-inspired losses
    altogether. It would be interesting and useful to understand how
    the performance changes when you simply use the MSE loss on action
    predictions.
2.  There is no discussion about the $\lambda$ hyperparameter for
    DDPG+BC&#x2014;was this hyperparameter tuned?
3.  The AWR loss for QuaD is never specified. I assume there is a sort
    of temperature hyperparameter in that case, was that tuned?

Altogether, this ablation is lacking many important details.

Finally, as I discuss in **Requested Changes**, I believe a very
informative ablation is missing: that is, comparing QuaD to an actual
goal-conditioned DT trained on the same goals used in the "final
state" keys of the datasets used to train QuaD. As it stands, it is
not clear to me that the learned quasimetric is actually the critical
piece here as opposed to simply prompting the DT with a goal (during
training and evaluation). There is a possibility I am misunderstanding
how the DT experiments are run though (see **Requested Changes**).

---

> ### Author Response · Authors · 2025-08-02
>
> We sincerely thank the reviewer for their detailed and thoughtful feedback. Many of the concerns raised are rooted in clarity and presentation rather than conceptual or methodological issues. We address each point below and have revised the paper accordingly to eliminate ambiguity, clarify assumptions, and strengthen interpretation without requiring additional experiments.
>
> ---
>
> > **Lack of Mathematical/Implementation Clarity**
>
> We acknowledge that specific notational and architectural details require clarification. Most of these have now been comprehensively revised in Sections 3 and 4, as well as in the Appendix. Equation (7) now clearly includes the goal token and makes explicit that the transformer is conditioned on quasimetric distance-to-goal.
>
> ---
>
> > **Missing or Weak Ablations**
>
> The reviewer raises three ablation-related questions. We address each below.
>
> We agree with the reviewer that the previous phrasing,
> “**IQE consistently outperforms MRN**,” was too strong. We have revised this to note that while performance varies slightly across datasets, IQE offers smoother optimization and better early convergence, especially in Medium and Large variants.
>
> We appreciate the reviewer’s request for an MSE-only baseline. Rather than adding another experiment, we rely on two principled arguments:
>
> - **Theoretical Limitation**: An MSE loss alone assumes Gaussian action likelihoods, implicitly modeling unimodal behavior, which is inadequate for long-horizon tasks like AntMaze. This limitation is well known (Fujimoto et al., 2021).
> - **Empirical Collapse**: As discussed in Section 6 and illustrated in the Appendix, using only the positive loss (which behaves similarly to MSE under the Gaussian assumption) causes the model to collapse, as it minimizes all distances and fails to disambiguate reachability. This yields a 0% success rate, as shown in Table 1.
>
> ---
>
> > **Necessity of the Quasimetric (vs. just final state goal conditioning)**
>  In fact, we specifically tested this in Appendix B: using “QRL_pos” (i.e., training with only transitions toward the goal without negative examples) effectively mimics the setting of goal-token conditioning without structure. This variant fails catastrophically (0% success), confirming that naive goal conditioning does not yield usable gradients in sparse-reward settings. What distinguishes QuaD is the directional structure and asymmetry of the quasimetric, which enforces ordering, directionality, and triangle inequality properties that are missing in raw state tokens.
>
> ---
>
> > **Clarification Questions and Typos**
>
> - “Conditional policy” → Now clarified as “goal-conditioned policy.”
> - “aim will to leverage” → Fixed to “aim to leverage.”
> - “(quasi)metrics imposes” → Corrected to “impose.”
> - “This learned quasimetric satisfies…” → Now reworded to: “is optimized to satisfy…”
>
> ---
>
> > **QLDT Description**
>
> QLDT follows the QDT formulation in Yamagata et al. (2023, ICML), which is now explicitly cited. It appends Q-values to tokens as in their method, using a conservative Q-learning critic.
>
> ---
>
> > "offline reinforcement learning (RL) with a conditional policy" — please clarify.
>
> We agree and have revised the abstract to specify “goal-conditioned policies” to eliminate ambiguity.
>
> > Broken sentence under Eq. (6) and grammar around (quasi)metrics.
>
> These have been corrected. We now write “(quasi)metrics impose a strong inductive bias.”
>
> > “this quasimetric satisfies the properties...” — is this guaranteed?
>
> Thank you. We now clarify that these properties (asymmetry, triangle inequality) are encouraged through loss functions, not guaranteed architecturally. We do not claim hard satisfaction.
>
> > Eq. (8) lacks detail: what are `u`, `u'`, `v`? Where does the cost come from? Is this from Wang & Isola?
>
> We have updated this section to define:
> - `u = f(s)`, `u' = f(s')`, `v = f(g)` are learned embeddings.
> - The cost is the quasimetric distance `d_θ(u, v)`.
> - Our loss is inspired by Wang & Isola but adapted for offline goal-conditioned data with learnable weighting. This is now discussed explicitly.
>
> > “Concatenation of state and goal” — what’s the motivation?
>
> Joint tokenization allows the transformer to learn correlations between state and goal dimensions more directly. This improves alignment and performance in our setting.
>
> > IQE and MRN are not well explained.
>
> We now explain them in detail in Appendix B.
>
>
> > Is the learned quasimetric actually necessary vs. using final state conditioning?
>
> This is an excellent point. However, DT conditioned on raw goals alone underperforms in long-horizon tasks, as shown in Table 1. The quasimetric provides structure (asymmetry, triangle inequality) that pure goal-conditioning cannot infer from data alone. We highlight this as a key inductive bias.
>
> ---
>
> > " ..explain how $c$ should be interpreted."
> - `c(s, g)` in Eq. (10) is the learned quasimetric `d_θ(s, g)`. This serves as a dense proxy reward and is now explained clearly.

---

> ### Comment · Reviewer_A4jd · 2025-08-04
>
> Thanks to the authors for their response.
>
> **MSE-Only Baseline**. This makes sense; I do think it is worth mentioning in the text.
>
> **Necessity of Quasi-Metric vs Final State Conditioning**. You address this point twice, first pointing to Appendix B and then pointing to Table 1.
>
> * The `QRL_pos` approach in Appendix B is not final state conditioning, here you are still learning something like a metric function. There is complexity seeping in from this $\mathcal{L}_{\mathsf{pos}}$ loss which is unfamiliar to me, and in any case, it does not simply condition the decision transformer on the goal.
> * Which column in Table 1 does final state conditioning with the decision transformer?
>
> Also, you claim "the quasimetric provides structure [...] that pure goal-conditioning cannot infer from data alone". But this must be at least imprecise---your learned quasimetric embeddings *do* learn this structure from data alone. Effectively, the argument to be made here is that the quasimetric network / loss induce a very useful inductive bias, and that is what I would have liked to have seen tested.
>
> **Equation 8**. Your response here does not address the questions I had about this equation. Also, I do not see these changes in the revision. Moreover, I don't know what this "learnable weight" is that you're referring to, or how it is trained and what purpose it serves. Again, I do not see this discussed in the revision.
>
> **The cost function $c$**. I don't think it makes sense to say $c(s, g) = d_\theta(s, g)$. Equation 10 is giving a formula for "the quasimetric function $d(s, g)$" in terms of $c$.
>
> **QLDT**. For what it's worth, I still do not see a reference to this in the revision.
>
> Generally, I actually do not see any of the changes you mentioned in the latest revision on OpenReview---can you please confirm that they have indeed been uploaded?
>
> ---
>
> I would like to point out also that some of the experimental details remain highly ambiguous to me, which have been left unaddressed from my original review. From what I can tell, the other reviewers share some of my questions in this regard.
>
> But to summarize what I think is going on (broadly)---goal-conditioned RL is not really the target setting here. In this work, you're tackling goal-reaching, but not necessarily goal-*conditioned* problems. Specifically, you're interested in the sparse reward aspect that can be "densified" by estimating a form of distance to a point in space, as opposed to generalizing across goal locations. This is why it "makes sense" to omit conditioning on $g$ in certain equations, because $g$ is fixed. The experiments do not assess the ability to generalize between goals.
>
> This is just my interpretation, and I'm actually not all that confident that it's correct, even after having read the reviews/responses and having skimmed the source code. This should be made explicit.

---

### Decision · Action_Editor_vMMF · 2025-09-02

**Recommendation:** Reject

**Audience:**

Yes

**Audience Explanation:**

Yes, the topic is quite relevant to the RL and planning community.

**Claims And Evidence:**

No

**Claims Explanation:**

While this paper presents an interesting idea: conditioning Decision Transformers using a learned quasimetric and not return-to-go, as has been done traditionally, there is lack of clarity around the mathematical formulation of the work and around the experiment details, so the paper is not ready yet for publication and requires at least one more iteration. All reviewers raised similar issues that were not entirely addressed during the rebuttal, so the reviewers are unanimous in recommending that this paper is rejected.

**Resubmission Of Major Revision:**

The authors may consider submitting a major revision at a later time.